# Audits Under Resource, Data, and Access Constraints: Scaling Laws For Less Discriminatory Alternatives

**Sarah H. Cen**
Stanford University
Palo Alto, CA 94304
shcen@stanford.edu

**Salil Goyal**
Stanford University
Palo Alto, CA 94304
salilg@stanford.edu

**Zaynah Javed**
Stanford University
Palo Alto, CA 94304
zjaved@stanford.edu

**Ananya Karthik**
Stanford University
Palo Alto, CA 94304
ananya23@stanford.edu

**Percy Liang**
Stanford University
Palo Alto, CA 94304
pliang@cs.stanford.edu

**Daniel E. Ho**
Stanford University
Palo Alto, CA 94304
deho@stanford.edu

## Abstract

AI audits play a critical role in AI accountability and safety. They are particularly salient in anti-discrimination law. Several areas of anti-discrimination law implicate what is known as the "less discriminatory alternative" (LDA) requirement, under which a protocol is defensible if no less discriminatory model that achieves comparable performance can be found with reasonable effort. Notably, the burden of proving an LDA exists typically falls on the claimant (the party alleging discrimination). This creates a significant hurdle in AI cases, as the claimant would seemingly need to *train* a less discriminatory yet high-performing model, a task requiring resources and expertise beyond most litigants. Moreover, developers often restrict access to their models and data as trade secrets, hindering replicability.

In this work, we present a procedure enabling claimants to determine if an LDA exists, even when they have limited compute, data, and model access. To illustrate our approach, we focus on the setting in which fairness is given by demographic parity and performance by binary cross-entropy loss. As our main result, we provide a novel *closed-form* upper bound for the loss-fairness Pareto frontier (PF). This expression is powerful because the claimant can use it to fit the PF in the "low-resource regime," then extrapolate the PF that applies to the (large) model being contested, all without training a single large model. The expression thus serves as a *scaling law* for loss-fairness PFs. To use this scaling law, the claimant would require a small subsample of the train/test data. Then, for a given compute budget, the claimant can fit the context-specific PF by training as few as 7 (small) models. We stress test our main result in simulations, finding that our scaling law applies even when the exact conditions of our theory do not hold.

## 1 Introduction

Complex AI systems are increasingly used in decision-making contexts that are subject to legal oversight. For example, HireVue has used AI to score video interviews [30]; various US agencies apply facial recognition for fraud detection [3, 4, 5]; and insurance providers use elaborate behavioral data to price policies [45, 46]. In the absence of methods that test whether such systems comply with the law, decisions that rely heavily on AI are permitted to escape scrutiny. Accordingly, there is growing need for AI audits. This need is particularly pronounced as models increase in complexity and scale. In particular, auditing becomes more challenging as models process high-dimensional,

unstructured inputs such as text and video, where the number of possible inputs becomes too large to test exhaustively. We henceforth refer to the system being audited as the "model," noting our analysis applies broadly, e.g., to human-AI decisions.[1]

AI audits are particularly salient in anti-discrimination law, in which the goal is typically to determine whether a protocol discriminates on the basis of a protected attribute. Several areas of anti-discrimination law implicate the **"less discriminatory alternatives" (LDA) requirement**. Conceptually, it holds that a protocol (e.g., AI model) should not be used if a protocol that achieves comparable performance while being less discriminatory can be found without causing "undue hardship" (i.e., imposing significant costs on the employer). For most of this work, we ground our discussion in a well-known context involving LDAs: Title VII of the US Civil Rights Act. Importantly, under Title VII, the burden of proving that an LDA exists *falls on the plaintiff*.

The LDA requirement highlights a fundamental and recurring issue in AI audits that is at the heart of our work: claimants generally have *fewer resources, less expertise, and limited knowledge* about the audited model than the defendants they challenge, and yet *bear the burden of proof*. We refer to this phenomenon as the "**resource-information asymmetry**" in AI evidence production.

This asymmetry can prevent claimants from gathering sufficient evidence to prove their claim. The LDA requirement drives this point home. Interpreted straightforwardly, the claimant is asked to *provide* a less discriminatory model, i.e., train a model that maintains the same level of performance as state-of-the-art, production models while simultaneously reducing discrimination. However, most claimants lack the compute, data, and expertise to do so. To add to the asymmetry, developers fiercely protect their models, training procedures, and data as trade secrets, meaning that claimants wishing to train a comparable model lack critical information.

## 1.1 Contributions

In this work, we tackle the resource-information asymmetry, using the LDA requirement as our case study. As is customary in disparate impact analyses, we focus on the classification setting (noting that this *includes* decisions that use generative AI to classify individuals). While our primary objective is to provide tools for claimants, we note that our approach *also helps model developers* assess the existence of LDAs without expending significant resources.

Conceptually, we first shift the problem from that of *training* a less discriminatory model to showing that a less discriminatory model *exists*, as explained in Section 2 and visualized in Figure 1a. We thus recast the problem to one of finding the performance-fairness Pareto frontier (PF). If the contested model sits far from the PF, then many LDAs exist. One could thus measure how easily the defendant could find an LDA using the distance from the PF, as formalized in Section 3. An additional benefit is that finding the PF does not reveal potentially proprietary information specific to the contested model. Note that, to use PFs, one must quantify performance and fairness; though not always feasible, this is rarely prohibitive because disparate impact analyses of AI generally involve quantitative evidence.

Recasting the problem in terms of PFs does not immediately resolve the resource-information asymmetry because finding the PF is a challenging problem in and of itself (see Figure 1b). Existing methods for finding the PF are just as (often *more*) resource-intensive than producing an LDA (see Sections 2 and 6). Our approach transforms this task into a tractable one by finding a scaling law for the PF, as visualized in Figure 2. Given the form of the scaling law, claimants could first fit its parameters in a low-resource environment, then extrapolate where the PF falls for the contested (large) model. Thus, the claimant would never need to train a large model.

This approach hinges on the existence of a scaling law. We provide, to our knowledge, the first *closed-form upper bound* of a performance-fairness PF (that is also, in a sense, tight). We describe how this expression can be used as a scaling law that allows one to learn the PF empirically at a fraction of the cost of existing methods. Our result uses binary cross-entropy loss as our performance metric, demographic parity as our fairness metric, and a posited data-generating process, as given in Sections 3 and 4. We hope to extend this result to other fairness and performance metrics in future work. As mentioned above and described in Section 4.2, to apply the scaling law, one needs four ingredients: (i) test data for evaluation, (ii) a small subsample of the training data, (iii) an estimate

---

[1]We adopt this terminology to simplify our language. Using "model" to refer to the broader AI system that results from pre-training, post-training, guardrailing, etc. is increasingly common.

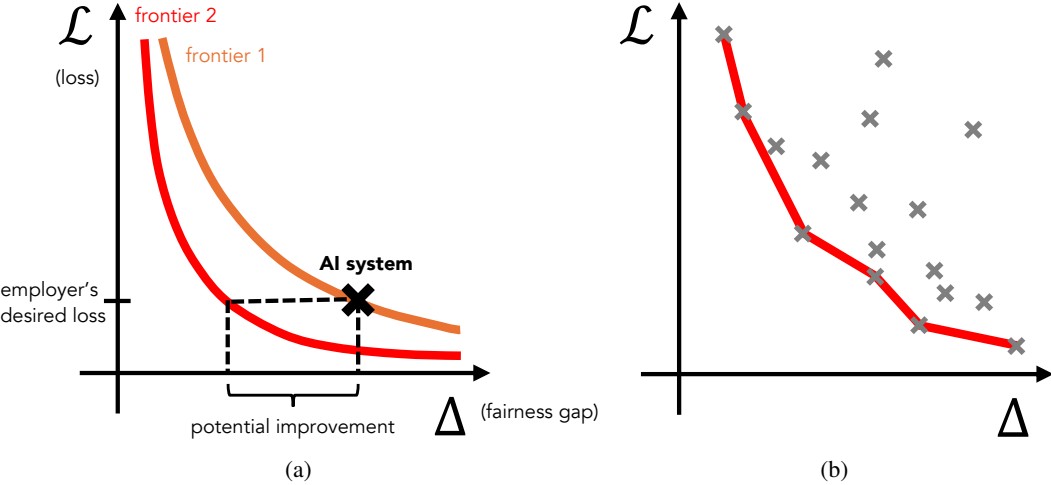

Figure 1: **(a)** Consider the contested AI model, as given by the "X". As described in Section 2, there are two steps in a disparate impact case before the less discriminatory alternatives (LDA) step. In the first step, the claimant establishes that there is disparate impact, typically using a statistical measure of discrimination (the fairness gap). Thus, the first step identifies the x-coordinate of the model (of "X"). In the second step, the defendant counters by arguing that the disparity serves what is known as business necessity, demonstrating that it allows them to meet some desired performance (the y-coordinate of "X"). In the final step, the claimant must prove that the employer could have used an LDA that would be less discriminatory while still meeting the employer's desired performance (i.e., a model that is to the left of but no higher than "X"). In this work, we cast this problem as determining where the performance-fairness Pareto frontier (PF) lies. For example, if the PF is given by the orange curve, then the "X" is on the PF, meaning that no LDAs exist; to decrease the fairness gap, one must sacrifice performance. If the PF is given by the red curve, there is significant room to improve fairness while maintaining performance. **(b)** Recasting the problem to finding the PF does not immediately resolve the resource-information asymmetry. The typical approach is to train many models while inducing them to have different loss-fairness characteristics (as given by the small x's), then trace the curve connecting the points lowest and to the left (red line). The issue is: for the PF to apply, the trained models must be of comparable size to the contested model. Training such models requires resources, data, and expertise beyond most claimants.

of the contested model's size, and (iv) an estimate of the amount of data used to train the contested model. Of these, (i) should not be difficult to obtain, as both parties use test data for the two steps of disparate impact cases preceding the LDA step (see Section 2). While the defendant must provide (ii)-(iv), they are more conservative asks than the status quo and may thus help strike a balance between the plaintiff's and defendant's interests.[2]

We conclude with experiments on synthetic data in Section 5 to stress test the conditions of our theoretical result. We find that our scaling law holds even when the conditions of our theory, outlined in Section 4, do not. We emphasize that our method is plug-and-play: as researchers develop better methods for finding Pareto-optimal models, our approach provides increasingly better estimates.

Our main contributions are summarized as follows:

1. We cast the problem of establishing the existence of an LDA and/or estimating the cost of producing an LDA as determining the performance-fairness Pareto frontier (PF).
2. We provide a closed-form upper bound of the performance-fairness PF that functions as a scaling law. We describe how one would apply this scaling law to determine whether an LDA exists at a fraction of the cost of training a large model. Our method works on unstructured inputs.
3. We conduct small-scale experiments to test our theoretical result.

Finally, we note that although we study LDAs, we believe that the need for low-resource, low-information methods to substantiate AI claims and conduct AI audits is a broad and pervasive issue. We point out extensions to our work in Appendices A and B.1.

---

[2]Straightforward approaches to finding an LDA require access to the trained model (e.g., to use it as a starting point from which one searches locally for an alternative) and/or a significant portion of its training data (e.g., to train a comparable model).

# 2 Less Discriminatory Alternatives and Their Relation To Pareto Frontiers

In this section, we discuss the relation between less discriminatory alternatives (LDAs) and Pareto frontiers (PFs), in the context of Title VII of the US Civil Rights Act (CRA). Our analysis will allow us to map the LDA requirement to a formal problem statement in Section 3. A more extensive background and related work can be found in Section 6.

## 2.1 Three steps of disparate impact cases

Title VII prohibits employment discrimination on the basis race, color, religion, sex, and national origin, which are referred to as "protected attributes." Although there are several ways to litigate under Title VII, the legal theory that is typically applied to automated decisions is the disparate impact doctrine [8, 33].[3] Courts generally adopt a three-part, burden-shifting test to adjudicate disparate impact claims:

1. The plaintiff must show that a facially neutral employment practice has *disparate impact* on a protected class, i.e., leads to systematically different outcomes for individuals in different groups. For automated systems, this is generally done by showing a statistical disparity in outcomes.

2. If the plaintiff establishes disparate impact, the defendant (e.g., employer or employment agency) can counter by showing that the practice is for "*business necessity*" reasons (i.e., performance). Similarly to the above, a quantitative notion of the business goal is often adopted.

3. If the defendant successfully claims business necessity, the burden of proof shifts back to the plaintiff, who must prove that there is an *LDA*: a protocol that the defendant could find without undue hardship and that would serve the same employment goal as in Step 2 while causing less disparate impact. If the plaintiff cannot do this, they lose the case.

Our work addresses **Step 3**. We assume Step 1 (which is studied extensively by the algorithmic fairness community [21, 63, 27, 18, 23]) and Step 2 are complete. See Section 6 for further discussion.

*Example* 2.1. To illustrate the challenges in finding an LDA, consider an AI hiring tool that uses large language and video models to screen applicants. Suppose that the plaintiff shows that this tool has disparate impact with respect to sex (Step 1). The employer counters by showing that the tool improves business outcomes due to hiring (Step 2). In Step 3, the plaintiff must establish that a tool that has less disparate impact but comparable business outcomes exists. Naive approaches to this include retraining a model from scratch (which would require resources, expertise, and data that most plaintiffs lack) or modifying the existing model (which would require access to the model that developers are unlikely to grant via trade secret protections). We expand on this example and the challenges that plaintiffs encounter in greater detail in Appendix B.2.

## 2.2 Relation between LDAs and Pareto frontiers

The example in Appendix B.2 highlights the difficulties of providing an LDA given the resource-information asymmetry. However, the LDA requirement does not necessarily imply that the plaintiff must train a new model. Instead, the plaintiff can prove, to a sufficient standard, that such an alternative *exists*.[4] In Figure 1a, we illustrate how proving the feasibility of an LDA can be cast as determining that a given model lies far from the performance-fairness PF. If the defendant's model lies above and to the right of the PF, then the gap quantifies the effort needed to find an LDA.

Although recasting the LDA requirement as a problem of finding the PF helps mitigate the **information asymmetry** problem—since finding the PF does not require knowledge specific to the defendant's system—computing the PF remains resource-intensive. In fact, the general method for

---

[3]The other well-known legal theory is disparate treatment, which applies when a decision-maker intentionally treats an individual differently based on their protected class. In the context of automated decision-making and AI systems, disparate impact is typically applied due to the difficulty of ascribing intent to algorithms or models [8]. Although disparate treatment is not generally applied to automated and AI systems, some have begun to contemplate its application due to the rise of "reasoning" models that seem to verbalize "intent." In addition, some believe algorithms that use protected attributes as inputs should be subject to a disparate treatment analysis.

[4]We cannot say for certain how courts will receive this evidence. Even if courts do not believe it fully satisfies the LDA requirement, it can be used to as an intermediate step that supports the plaintiff's requests for further discovery, including data and model access that they may be initially denied (see points 2 and 3 in Section 2.1).

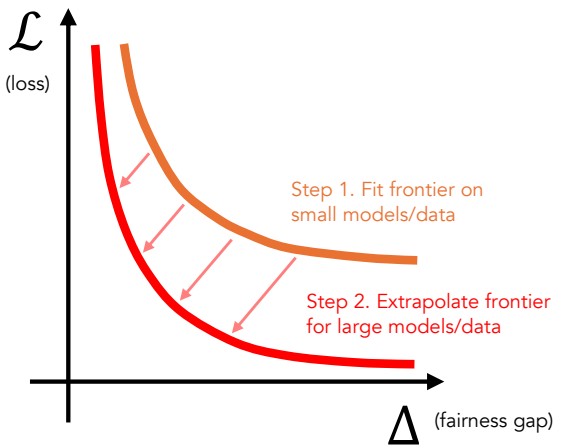

**Scaling law for loss-fairness PFs**

How does this approach address the resource-information asymmetry?

**Information asymmetry**. Claimants are often denied access to the model, data, and information about the training procedure. The PF is not model-specific and thus does not require knowledge about the contested model that may be considered proprietary.

**Resource asymmetry**. Current methods for finding a PF are costly (require training many large models → more resource-intensive than finding an LDA). Using a scaling law means the claimant never needs to train a large model.

Figure 2: An illustration of the proposed method on the left and how it addresses the resource-information asymmetry on the right, as explained in Section 2.

finding the PF is to train a wide array of models with varying performance-fairness characteristics, plot them on the performance-fairness plane, and then trace out the empirical PF [21, 61, 6], as illustrated in Figure 1b. From a resource perspective, this is *more* difficult than training a single large model—it involves training many!

Our main technical contribution, as outlined in Figure 2, is a scaling law that greatly reduces the resource cost of finding the PF. Our scaling law allows for several types of analyses, such as (i) estimating how far the contested model is from the PF for models of equivalent size trained on a similar amount of data; or (ii) estimating how many resources one would need to obtain a model with specific performance-fairness attributes.

## 3 Formal Problem Statement

In the remainder of this work, we demonstrate a method for proving the existence of a less discriminatory alternative (LDA) in a low-resource, low-information setting.

### 3.1 Setup

Consider a binary classification setting, where random variables $X \in \mathcal{X}$, $A \in \{0, 1\}$, and $Y \in \{0, 1\}$ denote the features, (binary) sensitive attribute, and (binary) class of an individual. Notably, $X$ may be unstructured. For instance, $X$ could denote a job applicant's resume, $A$ their sex, and $Y$ their suitability for the job of interest. Throughout, we use the convention that capital letters represent random variables, and their lower-case analogs denote specific values that they can take. Let $f : \mathcal{X} \times \{0, 1\} \to [0, 1]$ denote a (soft) classifier that takes in features $x$ and sensitive attribute $a$ and returns a score $f(x, a)$ between 0 and 1. Note that having $f$ depend on $a$ is without loss of generality, as $f$ could ignore it, so we use this $A$-"aware" setup. We use $\mathcal{F}$ to denote the model class (e.g., neural network of size $N$) that $f$ belongs to and $\mathcal{D}$ to denote $f$'s training dataset.

In this work, we use (expected) binary cross-entropy (BCE) loss as our measure of performance. Let the BCE of $f$ with respect to a (fixed) joint distribution $p_{X,A,Y}$ be given by

$$\mathcal{L}(p_{X,A,Y}, f) := -\mathbb{E}_{x,a,y \sim p_{X,A,Y}}[y \log(f(x, a)) + (1 - y) \log(1 - f(x, a))]. \quad (1)$$

We use demographic parity as our notion of fairness. Thus, the fairness gap of $f$ with respect to a conditional distribution $p_{X|A}$ is given by

$$\Delta(p_{X|A}, f) := \left| \mathbb{E}_{x \sim p_{X|A=1}}[f(x, 1)] - \mathbb{E}_{x \sim p_{X|A=0}}[f(x, 0)] \right|.$$

We refer to $p$ as the "test" distribution. Note that the plaintiff may choose to use other notions of performance and fairness. In this work, we use BCE and demographic parity, leaving an analysis of

other definitions to future work.[5] The "right" choice of fairness metric is generally context-dependent and highly debated. We provide a short discussion in Appendix B.3. Note that history indicates that demographic parity is a likely choice by courts, as selection rates have appeared across multiple contexts, including the Four-Fifths Rule from 1978 to NYC's Local Law 144 of 2021.

### 3.2 Our objective: Establishing existence of an LDA with limited information and resources

*Limited resources.* Suppose the plaintiff's model belongs to a model class $\mathcal{F}^-$ and is trained on a dataset $\mathcal{D}^-$, which are much smaller than the model class $\mathcal{F}^+$ and training dataset $\mathcal{D}^+$ available to the defendant. Let $N^+$ and $D^+$ denote the parameter count of $\mathcal{F}^+$ and number of samples in $\mathcal{D}^+$. Similarly, let $N^-$ and $D^-$ denote the parameter count of $\mathcal{F}^-$ and number of samples in $\mathcal{D}^-$. Suppose that the claimant can only train models that belong to $\mathcal{F}^-$ and $\mathcal{D}^-$, where $N^+ \gg N^-$ and $D^+ \gg D^-$. We assume that $\mathcal{D}^-$ is drawn from the same distribution as $\mathcal{D}^+$.

*Limited information.* In past and ongoing cases, judges and defendants have been reluctant to share information about the model and training data, arguing that this information is proprietary and confidential. In this work, we assume the claimant is similarly denied access to the contested model but can request a limited amount of information. We propose that, since the LDA requirement asks that the claimant assess what alternatives are "feasible" or "reasonable" for a defendant to obtain, the claimant should minimally be given information about how many resources were used to train the contested model: namely, the values $N^+$ and $D^+$. We further propose that the claimant should be granted a small subsample of training and test data. In the notation given above, the subsample of training data would be of size $D^-$, and the subsample of test data at most $D^-$.[6]

*Objective: Show existence of a (feasible) LDA.* Our objective is to provide a method that the claimant can use to determine whether the contested model is at least $\delta$-far from the Pareto frontier (PF) for a fixed performance level (i.e., loss), which would imply that finding an LDA is $\delta$-feasible.

Formally, let the defendant's contested model be denoted by $\hat{f}^*$. Our objective is to provide a method that the claimant can use to determine whether, for a given value $\delta > 0$, there exists a model $\hat{f}$ of size $N^+$ and trained on a dataset of size $D^+$ such that

$$\mathcal{L}(p_{X,A,Y}, \hat{f}) \leq \mathcal{L}(p_{X,A,Y}, \hat{f}^*) \quad \text{and} \quad \Delta(p_{X|A}, \hat{f}) < \Delta(p_{X|A}, \hat{f}^*) - \delta. \tag{2}$$

In other words, there exists a model that is at least as performant as the contested model and reduces the contested model's fairness gap by at least $\delta > 0$. The larger the value of $\delta$, the greater the distance between the contested model and the PF. This distance is indicative of the "feasibility" of finding an LDA: under the mild assumption that the loss-fairness PF monotonically improves as $N$ and $D$ increase, large values of $\delta$ indicate that the defendant could have found an LDA with a small fraction of the resources they expended to train $\hat{f}^*$. Our scaling law also allows us to estimate the minimal $N$ and $D$ needed to achieve certain loss-fairness characteristics.

## 4 Main Result: Scaling Law for Loss-Fairness Pareto Frontier

In this section, we derive a scaling law for the loss-fairness Pareto frontier (PF) under a given data generating process (DGP). We then describe how one applies this scaling law. Note that we present an intermediate result in Appendix G that lends intuition for how our main result is obtained and can be used as a building block for future works that examine different notions of fairness and DGPs. All proofs are given in Appendices H and I.

### 4.1 Closed-form upper bound of the loss-fairness Pareto frontier

Recall that a Pareto frontier (PF) is traced out by the lowest-loss classifier for every fairness gap value $\underline{\Delta}$. In this section, we present our main result: a closed-form upper bound of the loss-fairness PF. Before presenting the result, we introduce some notation. Let $p$ and $q$ denote joint distributions over random variables $X$, $A$, and $Y$. Let $p(1 \mid x, a)$ and $q(1 \mid x, a)$ denote the Bayes optimal classifiers under $p$ and $q$, respectively. We begin with an assumption that we explain in Appendix C.1.

---

[5]Importantly, recall from Section 2 that the plaintiff chooses their definition of disparate impact in Step 1 and the defendant chooses their definition of business necessity/legitimate employment goal in Step 2.

[6] This amount of information is even more conservative than what has been requested in past cases [2, 49] and what practitioners currently advocate for [16, 17].

**Assumption 4.1** (Model misspecification symmetry). *For a given distribution $q$, let $\hat{f}$ be the loss-minimizing classifier in model class $\mathcal{F}$ when training on a dataset $\mathcal{D}$ drawn from $q$. We assume that $\mathcal{L}(q_{XAY}, \hat{f}) - \mathcal{L}(q_{XAY}, q_{1|X,A})$ is a constant $c_1(\mathcal{F}, D)$, that does not depend on $q$ or $\hat{f}$. Moreover, $\mathbb{E}[\hat{f}(x, A)|A = 0] - \mathbb{E}[q(1|x, A)|A = 0] = \mathbb{E}[\hat{f}(x, A)|A = 1] - \mathbb{E}[q(1|x, A)|A = 1]$, where all expectations are taken with respect to the test distribution $p_{X|A}$.*

This assumption essentially implies that each Pareto-optimal model $\hat{f}$ that belongs to model class $\mathcal{F}$ is symmetric with respect to the DGP, which one would expect to hold for large models that serve as universal function approximators. The second condition in Assumption 4.1 implies that $\hat{f}$ is symmetric with respect to $A$. This condition does *not* imply that $\hat{f}$ fits group $A = 0$ as well as the other group $A = 1$, but that the "distance" between $\hat{f}$ and $q(1 \mid x, a)$ is symmetric with respect to $A$.

**Theorem 4.2.** *Consider the following data generating process: $A \sim Ber(\pi)$, $X \perp A$, and $Y|X, A \sim Ber(\sigma(g(X) - \zeta A))$ for $\zeta \geq 0$, where $\sigma(\cdot)$ is the sigmoid function, $g$ is a measurable function, and $q$ is the joint distribution induced by some $\zeta$. With slight abuse of notation, let $\underline{\Delta} := \Delta(p_{X|A}, \hat{f})$ denote the fairness gap of $\hat{f}$ under test distribution $p$. We assume $p = q^{\zeta_p}$ for a fixed $\zeta^p$ and all Pareto-optimal $\hat{f}$ satisfy Assumption 4.1 for some $q^\zeta$. Then, for all $\hat{f}$ on the Pareto frontier,*

$$\mathcal{L}(p_{X,A,Y}, \hat{f}) \leq B(\mathcal{F}, D) - c \cdot c' \log(1 - c'' + \underline{\Delta}) - c \cdot (1 - c') \log(c'' - \underline{\Delta}) \tag{3}$$

$$+ (c'' - \underline{\Delta})(1 - c'' + \underline{\Delta}) \left( \frac{c \cdot c'}{2(1 - c'' + \underline{\Delta})^2} + \frac{c \cdot (1 - c')}{2(c'' - \underline{\Delta})^2} \right) + \varepsilon,$$

*where $\varepsilon = \mathcal{O}\left( \max_\zeta \mathbb{E}_{p_{X|A=1}} \left[ (q^\zeta_{Y|X,A}(1 \mid x, 1) - \bar{q}^\zeta_1)^3 \right] \right)$ and $\bar{q}^\zeta_1 := \mathbb{E}_{p_{X|A=1}} [q^\zeta_{Y|X,A}(1|x, 1)]$. $B(\mathcal{F}, D)$ is a constant that depends on the model class $\mathcal{F}$, dataset size $D$ and $p$, and $c, c', c'' \in [0, 1]$ are constants that depend only on $p$.*

This result expresses the loss of each Pareto-optimal $\hat{f}$ on $p$ as a function of $\hat{f}$'s fairness gap $\underline{\Delta}$. The dependence of loss on fairness gap is fully characterized, except for four constants that do not depend on $\underline{\Delta}$: $B(\mathcal{F}, D)$, $c$, $c'$, and $c''$. The constants $c$, $c'$, and $c''$ depend only on $p$. $B(\mathcal{F}, D)$ depends on $p$, the model class $\mathcal{F}$, and dataset size $D$. We describe how this result can be used as a scaling law in Section 4.2. Finally, we note that although our result is specific to the DGP given in Theorem 4.2, we hope it provides a template for future analyses. Even so, the DGP above is fairly general: that $A$ is Bernoulli is without loss of generality; and $X \perp A$ does not hold in general but should not affect a Pareto-optimality analysis.

## 4.2 Applying the Scaling Law to the LDA Requirement

Theorem 4.2 can be viewed as a scaling law. Indeed, it provides a closed-form expression for the loss-fairness Pareto frontier (PF) that involves three constants that do not depend on $\mathcal{F}$ and $D$, then one constant $B(\mathcal{F}, D)$ that does. This implies that finding the PF for a large model class $\mathcal{F}^+$ and large dataset size $D^+$ involves three main steps, as follows.

To apply the scaling law, the auditor needs an estimate of $N^+$ and $D^+$, a subset of the train/test data, and a compute budget, as described in Section 3. As a first step, the auditor trains models of size $N^- \ll N^+$ on a dataset of size $D^- \ll D^+$, where $N^-$ and $D^-$ are chosen based on the auditor's compute budget. These trained models yield an empirical PF that can be used to fit a curve of the form given in Theorem 4.2, i.e., estimate the unknown constants in the expression. The auditor can repeat this process with different $N^-$ and $D^-$ until they are satisfied with their fit. Then, given an expression for how $B(\mathcal{F}, D)$ changes with $\mathcal{F}$ and $D$, one can plug $\mathcal{F}^+$ and $D^+$ into $B(\mathcal{F}^+, D^+)$ to extrapolate the PF for $\mathcal{F}^+$ and $D^+$. Luckily, past works provide a template for the form of $B$; specifically, previous works such as [32, 7] show that the minimum loss scales with $\Theta(1/N^\alpha + 1/D^\beta)$ for some $\alpha, \beta > 0$. Because $B$ is a linearly additive constant in Theorem 4.2, this implies that $B$ also scales at the same rate. A more detailed, step-by-step procedure is given in Appendix D.

We provide a discussion of limitations and corresponding future work in Appendix A.

*Remark.* The existence of a closed-form expression for the PF does *not* mean that all loss-fairness PFs look the same. In fact, the shape of the PF can vary significantly depending on the values of the constants, as we show in Appendix J.1. The constants are *context-dependent*, and thus the shape of the PF is also *context-dependent*. We also note that our approach "fails gracefully" in that it is *conservative*: if a claimant finds the contested model is $\delta$-far from the PF, then it is *at least* that far. See Appendix D for further discussion.

# 5 Experiments

We run both simulations and experiments to test our main result. The simulations can be found in Appendix J.1. In this section, we present experiments that we ran to test Theorem 4.2 and its assumptions. We find that the PFs predicted by our theory closely matches the empirical PFs, validating our scaling law approach to assessing the existence of a less discriminatory model (LDA). Our experiments also demonstrate how one might apply Theorem 4.2 in practice.

**Main goals.** These experiments allow us to manipulate key aspects of the data generating process and training setup and stress test the conditions of Theorem 4.2 in three critical ways:

1. We relax the assumption on the data generating process in Theorem 4.2 so that $X \not\perp A$.

2. One of our main proof techniques is to transform the constrained minimization problem into an unconstrained one by creating an artificial train distribution $q$ (that does not actually exist). Doing so allows us to "tune" the fairness gap, then find the loss-minimizing model. In our experiments, we test whether this approach is representative of the true loss-minimizing models at various fairness gap values.

3. Our second proof strategy is to use a Bayes optimal estimator for an artificial distribution to approximate a Pareto-optimal model. This approach further relies on Assumption 4.1. We trace out the empirical PF by training a collection of models on the synthetic train data, using the lower convex hull of the resulting (loss, fairness) characteristics as the empirical PF. Thus, our experiments test whether the approach we take to derive our analytic result holds empirically.

**Synthetic data generation.** We generated a synthetic dataset with 10,000 samples, where each sample is a 20-dimensional vector with binary group labels. Each $x$ is sampled i.i.d. from a standard isotropic multivariate Gaussian distribution. The group label $a \in \{0, 1\}$ is deterministically assigned by linearly projecting the first two dimensions of $x$ onto a scalar, then thresholding, where the linear weights and bias are randomly sampled from a multivariate Gaussian, and the threshold is chosen so that a fixed fraction of the population belongs to Group 1 (i.e., $p(A = 1)$ is the desired value). Because $A$ depends on $X$, they are not independent, as required in Theorem 4.2; our experiments therefore test whether the closed-form PF holds despite this relaxation.

The target variable $y \in \{0, 1\}$ is sampled from a Bernoulli distribution with probability $\mathbb{P}(Y = 1 \mid X, A) = \sigma(g(X) - \zeta A)$, where $g : \mathbb{R}^{20} \to \mathbb{R}$ is a fixed, randomly initialized 2-layer MLP with a hidden layer of size 32 applied to $X$, $\zeta \geq 0$, and $\sigma(\cdot)$ denotes the sigmoid function. It thus mirrors the data generating process in Theorem 4.2, except that $X$ and $A$ are not independent. We show results for various values of $\zeta$. Finally, our experiments use the same distribution for the train and test sets.

**Obtaining the empirical PF.** We obtain the empirical PF by training models with varying loss-fairness characteristics. To do so, we use linear scalarization, i.e., we set the loss to be the sum of the BCE loss and the fairness gap $\mathcal{L}_{\text{total}} = \mathcal{L}_{\text{BCE}} + \lambda \cdot \mathcal{L}_{\text{DP}}$ and vary $\lambda$.

As in our theoretical result, we use demographic parity as our fairness notion. We vary $\lambda$ across 100 values between -3 and 5, with a higher density of values in the -3 to 0 portion.[7] For every scalarization value, we run 3 trials, then trace out the empirical PF (with mild smoothing and averaging). Further training details are given in Appendix J. Note that one may choose to use a better method for obtaining the empirical PF, and hence need to both train fewer models and obtain a more accurate PF.

To get the empirical PF, we compute each trained model's BCE loss and fairness gap on the held-out test data. We plot these (loss, fairness) pairs on the BCE loss vs. demographic parity gap plane. To trace the empirical PF (the lowest achieved loss for each given demographic parity gap), we connect the points forming the lower convex hull of the points.

**Testing the scaling law.** To test whether the scaling law holds, we demonstrate that there is a fixed of constants $C_1$ through $C_7$ such that the closed-form expression given in Theorem 4.2 predicts the PF across different values of $N$ and $D$, as laid out in Section 4.2. We focus on the effect of model scaling, leaving $D$ constant and only varying the model size $N$. We then use the procedure described

---

[7] Although $\lambda$'s are usually positive, we use negative values to test whether the PF curves upward for large fairness gaps as predicted by our theory.

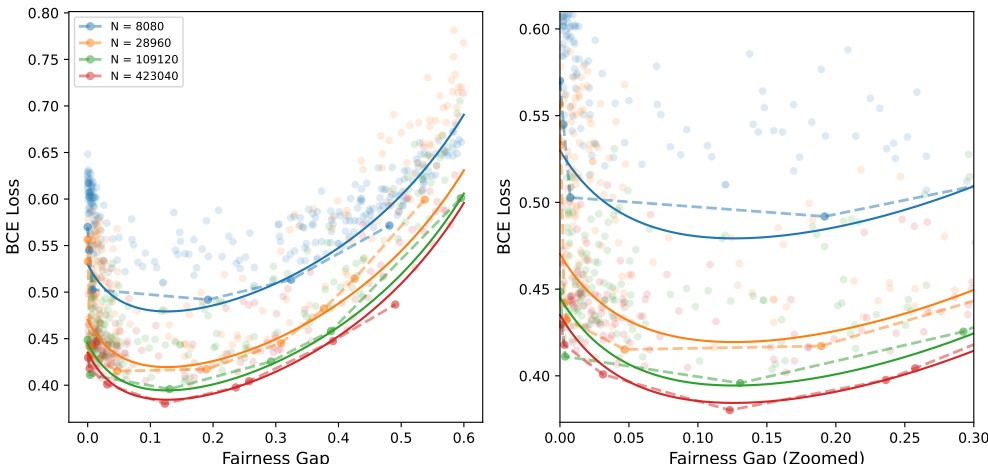

Figure 3: Pareto frontier for different model sizes under ($\pi = 0.2$) and bias strength $\zeta = 0.5$. Each point corresponds to a trained model with a different fairness regularization weight $\lambda$. The dashed lines show the empirical Pareto frontier, created by finding the lower convex hull of all the points. The solid lines show fitted curves to the points on the Pareto frontier using Theorem 4.2 with $c = 0.0176$, $c' = 0.92$, $c'' = 0.1424$, with bias $-0.285$, loss scaling law $55(N^{-0.7} + D^{-0.5})$. The left panel shows the frontier across the range of $\Delta$, while the right panel zooms in on $\Delta \in [0, 0.3]$. The fitted curve mimics the empirical data well though it is imperfect. We found that there were many possible fits, depending on the precise choice. We show another possible fit in Figure 11 below.

in Section 4.2 to "scale" the curve. *If the empirical PFs are close to the predicted curves, it would validate our method: that training small models can be used to approximate the PF of larger models.* In Figure 3 and similar plots in Appendix J, the dashed lines give the empirical PFs, and the solid lines give the ones predicted by our theory.

**Results.** Our findings corroborate our theoretical result: First, from the y-intercepts alone, we verify that $B(\mathcal{F}, D)$ nearly perfectly scales proportionally to $N^{-\alpha} + D^{-\beta}$ plus an additive constant, as predicted. Second, the empirical PFs follow the shape as our closed-form scaling law. Even small changes to the expression in Theorem 4.2 would not yield a similarly good fit. Perhaps somewhat surprisingly, our theory predicts that the effect of scaling via $B$ is linearly additive, and our experiments validate this prediction. Both of these findings provide strong evidence supporting: (i) our Assumption 4.1 and (ii) the use of Bayes optimal estimators on artificially constructed train distributions to approximate Pareto-optimal models (which allows us to obtain our result with minimal assumptions on the model class and for unstructured inputs).

We note that the fitted curve mimics the empirical data well though imperfectly, which may occur for several reasons. The first is that the empirical PF is always imperfect since finding Pareto-optimal models is difficult, e.g., due to factors discussed in Section 4.2 under "Considerations for Step 1." The second is that (3) is an upper bound—therefore, there is a possibility that the shape of the true PF differs slightly from the predicted one (though our result is, in some sense, tight as seen in Appendix I). The third is that there are many possible fits for the same empirical PF, as we show in Appendix J. In practice, one may choose to fit the curve as closely as possible to the data or, noting that the PF is an idealized concept, fit the curve to lower bound the empirical points.

## 6 Background and Related Work

**AI audits.** The study of AI audits has grown rapidly in the past few years [50, 44, 25, 51]. Although the range of methods and objectives for audits varies greatly, many agree that audits serve an important function in AI accountability, whether conducted by civil society groups, private actors, or regulators [14, 38, 48]. Researchers have identified multiple challenges to conducting AI audits [25], including the limited access granted to third-party auditors [57, 17, 16], conflicts of interest that can arise due to financial ties between auditors and auditees [20, 59], poor researcher protections [42], and difficulties in verifying the veracity of self-reported information [58, 22, 60]. Among these issues, our work seeks

to address the difficulties of producing (convincing) evidence in resource-, data-, and information-constrained settings. Specifically, the auditor (or claimant) is often at a disadvantage when producing evidence that surpasses the relevant (context-dependent) standard in that (i) they typically possess fewer resources than the other party (e.g., model developer), (ii) they have limited access to the system they seek to audit, and (iii) they may lack technical expertise or domain knowledge.

**Disparate impact doctrine.** In the US, disparate impact is a legal doctrine used to demonstrate that a facially neutral practice or practice discriminates against a protected group by showing that it has a disproportionately negative impact on them, even if there is no intent to discriminate [12]. It is often contrasted with disparate treatment, which concerns intentional discrimination [8]. Under disparate impact doctrine, plaintiffs typically show that a practice has disparate impact by providing statistical evidence. This doctrine was one of the main drivers behind the field of algorithmic fairness and the development of outcome-based fairness metrics [27].

There is precedent for assessing disparate impact using a guideline known as the Four-Fifths Rule, which states that a practice is has disparate impact if the selection rate for a protected group is less than four-fifths of the selection rate for the group with the highest rate. Although the use of selection rates to assess discrimination is highly debated (as are fairness metrics, more broadly), selection rates are often used in disparate impact analyses because they provide a simple and intuitive way to compare group outcomes that does not suffer from a selection bias problem (that makes the estimation of true positive and false positive rates difficult). Selection rates remain a popular choice in practice, e.g., see NYC's Local Law 144 of 2021.

**Less discriminatory alternatives (LDAs).** The LDA requirement arose in US anti-discrimination law, as described further in Section 2. It has become a critical part of disparate impact doctrine following the landmark *Griggs v. Duke Power Co.* decision in which the Supreme Court held that discrimination can occur even in the absence of intent (i.e., facially neutral practices can also be discriminatory) [12]. Within disparate impact doctrine, the LDA requirement urges decision makers to consider less harmful alternatives among equally effective options as a structural way to assess and remove unnecessary discrimination [9]. We refer readers to Lamber [39], not [1], Rutherglen [54] for further analyses of the role of LDAs in disparate impact claims. We further discuss this in Appendix E.

**Less discriminatory *algorithms*.** Recent works examine the LDA requirement as it applies to algorithmic decision-making. Gillis et al. [24] cast the search for LDAs as an mixed integer program, then directly search for an algorithm that minimizes error rate disparity subject to performance criteria. Laufer et al. [40] provide theoretical insights showing, among other takeaways, that identifying the least discriminatory alternative is NP-hard and may only constitute an LDA for the distribution or data over which it was derived. Our work mirrors the motivation of Gillis et al. [24] in that we seek to provide an operationalizable method for plaintiffs. However, as both Gillis et al. [24] and Laufer et al. [40] note, *producing* an LDA is computationally expensive and thus existing approaches do not scale to for large models, which are increasingly important to audit. Our work adds to this space by providing a tool for plaintiffs with limited compute and data.

**Performance-fairness Pareto frontiers (PFs).** In many settings a trade-off between fairness and performance emerges [10, 43, 37]. In such cases, the performance-fairness PF characterizes the best fairness and performance that can be simultaneously achieved (i.e., the best performance than can be achieved for a given fairness level, or vice versa). Various works propose methods for improving Pareto optimality or tracing out the performance-fairness PF, including Navon et al. [47], Ruchte and Grabocka [53], Singh et al. [56], Rothblum and Yona [52], Kamani et al. [34], Liu and Vicente [41], Chzhen and Schreuder [19], Zeng et al. [62]. In many of these works, the goal is to develop efficient methods for tracing out the PF, but they still assume the resources needed to train *at least* one model. In our work, we seek to improve scalability in a different sense: one can trace out the PF in small-model, low-data regimes and extrapolate the PF to large-model, large-data regimes (and thus one does not even need the resources required to train a large model). To the best of our knowledge, none of the existing works provide closed-form expressions for the fairness PF.

**Scaling laws.** Scaling laws have been widely studied in the context of deep learning, typically used to predict how model behavior improves with increasing model size, data, and compute [31, 36, 32, 7]. In this work, we seek to provide a theoretically-grounded scaling law for the loss-fairness PF that allows one to fit the PF with small models and small datasets, then extrapolate to larger models and datasets with the help of the power law (that applies to loss) found in the works above.

## Acknowledgments and Disclosure of Funding

We thank members of Percy Liang's group and Daniel E. Ho's RegLab for their valuable feedback during the early stages of this project. This work is supported by the CSET Foundational Research Grant on audit methods for frontier models.

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

# A    Conclusion, Limitations, and Future Work

In this work, we tackle the problem of resource-information asymmetry in AI audits and legal cases involving AI. We focus our discussion on Title VII's "less discriminatory alternative" (LDA) requirement as a case study exhibiting how claimants often face a steep uphill battle in order to meet their evidentiary burden (as noted in previous works [11, 24, 40]). We are motivated by this issue to develop tools and methods to reduce the hurdles claimants face. Our main contributions are (1) to cast the problem of finding an LDA as one of estimating the performance-fairness Pareto frontier (PF), (2) to provide a novel technical result that, to our knowledge, provides a first closed-form expression for the performance-fairness PF, and (3) to show how this result can be used as a scaling law for performance-fairness PFs that directly addresses both the resource and information asymmetry issues posed by the LDA requirement.

Next, we identify the assumptions (both technical and conceptual) of our work, which highlight potential limitations and avenues for future work:

1. Our theoretical analysis is conducted for specific notions of fairness and performance. We justify these choices in Section 3 and Section 6, and we believe that future work tackling other definitions would be valuable. Our work also relies on disparate impact and business necessity being measurable; we do not preclude the use of multiple metrics to measure fairness or performance, and studying the more complex multi-objective optimization problem in which there are more than two metrics of interest would be compelling future work. We are unsure how to address settings in which fairness and performance are not measurable, and we welcome future work that explores the LDA requirement in such settings (e.g., when fairness is ordinal).

2. Our result applies for the DGP given in Section 4.1. A compelling direction for future work would be to determine how well our result holds, for real data and *even* for other DGPs. Although stylized, the DGP we study still allows for significant generality. We also recommend future work, both theoretical and empirical, on the form of the loss-performance PF for other DGPs.

3. Relatedly, our main proof technique that allows us to obtain an analytic expression without strong assumptions on the inputs or the model class is to use a notion of "duality" that approximates Pareto-optimal classifiers with varying fairness characteristics using Bayes optimal estimators on artificially constructed training distributions. To do so, use $\zeta$ to "tilt" the (artificial) train distribution. As one explores other DGPs, one could also explore (i) the conditions under which this duality holds and (ii) alternative ways of tilting the training distribution.

4. Assumption 4.1 appears to hold in our experiments. Given that this assumption does not directly depend on the experimental choice of data generating process (DGP), this is strong evidence supporting it. However, we suggest two directions of exploration: (i) further theoretical work to understand when this assumption may not hold, which may result in an additional term in (3) that allows Assumption 4.1 to be removed and therefore strengthens the result; and (ii) empirical investigations to identify when this assumption breaks down in practice.

5. Our result (3) is an upper bound. Although it is tight in some sense (that it holds with equality for some $g$), we believe a compelling direction for future work is to identify a tighter bound.

6. As mentioned in Section 4.2, the form of $B$ that we adopt is borrowed from the literature on large language models. One could explore alternative forms of $B$ that may be more appropriate for other model classes and sizes.

We identify several other considerations and extensions:

1. A defendant may be able to argue that, in failing to produce a specific less discriminatory alternative (LDA), this approach does not pass the necessary evidentiary standard. As noted in Footnote 4, if the court does not consider this evidence strong enough to satisfy the LDA requirement, we hope that it can be used to support the plaintiff's requests for further discovery, including data and model access that they may be initially denied.

2. Our experimental results are limited. There are several directions for future work, including conducting experiments on real datasets, further stress testing the limitations of our theoretical results, and running experiments at greater scale.

3. Our approach requires access to some training and test data. We feel that this requirement is unavoidable, and our contribution is to significantly decrease the amount of data that the

claimant needs. However, we acknowledge that this requirement may still pose a hurdle and leave the exploration of techniques that further mitigate the data requirements to future work.

4. The empirical PF is, in a sense, a random variable that depends on the sampled training data and the randomness of the training procedure. One could explore the confidence intervals of the estimated PF that result from different training runs. Similarly, our theoretical result does not have a notion of uncertainty; future work could explore a high-probability version of our result.

# B  More Information on LDAs

## B.1  LDAs Beyond Employment Law

The "less ___ alternative" requirement expands beyond employment to areas including housing, lending, disability, and even environmental justice. The Department of Housing and Urban Development (HUD) has a 2023 rule returning to the 2013 Fair Housing Act standard under which actuarially sound housing-insurance policies are unlawful if an equally effective, less discriminatory practice is available. The Consumer Financial Protection Bureau's (CFPB) 2023 Fair Lending Report requires lenders to proactively search their credit-scoring models for LDAs even in the absence of litigation. A closely related duty appears in disability law; Equal Employment Opportunity Commission (EEOC) guidance under the American Disabilities Act (ADA) obliges employers to adopt any "reasonable accommodation" that meets business needs without exclusion, effectively functioning as an LDA requirement. In environmental justice, permits to proposed projects may be denied if a less harmful to human health and the environment exists (an "environmentally preferable alternative") under National Environmental Policy Act (NEPA).

Beyond the US, comparable concepts exist. The EU employs the principle of proportionality in discrimination cases, requiring that measures be appropriate and necessary, but also that the respondent show "there is no practicable alternative." One may argue that the LDA requirement parallels the EU's "data minimisation" requirements under the General Data Protection Regulation (GDPR), requiring data processing to be "limited to what is necessary." In Canada, the Meiorin test examines whether the employer has accommodated affected groups to the point of undue hardship. South Africa, drawing from Section 36 of its Constitution, applies a limitations analysis that considers whether less restrictive means could achieve policy objectives without discriminatory impacts.

## B.2  Example: Challenges of finding an LDA

To illustrate the challenges involved in finding an LDA, consider the following example. Consider an AI tool that uses large language and video models to screen application packages, which contain unstructured text and video interviews. Suppose that the plaintiff successfully establishes that the tool disproportionately favors candidates from a specific demographic group using one or more chosen metrics, such as the demographic parity gap (Step 1). Suppose further that the defendant successfully shows that the tool improves their business outcomes by reducing the amount of time to screen applicants while surfacing candidates that are well suited for each job (Step 2).

At this point, the plaintiff can only win the case if they are able to meet the LDA requirement (Step 3). They meet several hurdles in their attempts:

1. The plaintiff first attempts to train their own LDA. They quickly realize that training a comparable model to the defendant's production, state-of-the-art model requires significant compute.

2. Even if they are able to obtain the necessary compute, training a model requires access to good data. The plaintiff learns that collecting enough training data is prohibitively expensive, so the plaintiff requests it from the defendant (or the developer if the data is owned by a third party). The defendant and/or developer claims that sharing their training data is tantamount to releasing trade secrets and compromises user privacy, and the judge agrees.[8]

3. The plaintiff turns to a third option. They request access to the contested model so that they can use it a starting point to locally search for an LDA by, e.g., fine-tuning or probing its internal

---

[8]When the owner of the information is a third party that is not a direct party to the lawsuit (e.g., the plaintiff sues an employer, who outsources the AI model development to a third party), it can be even more difficult to obtain this information. Further discussion in Footnote 6.

representations. The defendant and/or developer claims that this, too, violates trade secrecy, and the judge agrees.

4. Finally, the plaintiff considers establishing the existence of an LDA by characterizing the range of possible classifiers, in the same spirit as analyses in [24, 40], but discovers these approaches work well on finite-dimensional, categorical inputs but not on unstructured ones.

At this point, the plaintiff may throw in the towel, conceding that they cannot meet the LDA requirement because the burden of providing an LDA is too high, especially given the limited resources, data, and model access that they possess.

### B.3   On the choice of fairness metric

In Section 3 we note that we choose demographic parity as our fairness metric, but in general this choice is context-dependent and debated upon. History indicates that demographic parity is a likely choice by courts, as selection rates (the backbone of the demographic parity metric) have appeared across multiple contexts, including the Four-Fifths Rule from 1978 to NYC's Local Law 144 of 2021. Moreover, selection rates do not suffer from the same selection bias issues as other metrics (see Section 6 for further discussion). Similarly, we study BCE loss because it is widely used as the performance metric for classifiers and is precisely what developers/companies optimize in training.

## C   Intuitions for Theoretical Assumptions and Results

### C.1   Intuition for Assumption 4.1

This assumption essentially implies that each Pareto-optimal model $\hat{f}$ that belongs to model class $\mathcal{F}$ is symmetric with respect to the DGP, which one would expect to hold for large models that serve as universal function approximators. That is, the "distance" between the loss-minimizing $\hat{f}$ in $\mathcal{F}$ learned on $q$ and the Bayes optimal classifier $q(1|x, a)$ may depend on the model class $\mathcal{F}$ and the number of samples $D$ it is trained on, but not on $q$. As discussed in the previous section, this may not always hold exactly (e.g., when some $q$'s are trivial to learn), but we believe it to be reasonable for deep learning models trained on sufficiently sophisticated tasks. The second condition in Assumption 4.1 implies that $\hat{f}$ is symmetric with respect to $A$. This condition does *not* imply that $\hat{f}$ fits group $A = 0$ as well as the other group $A = 1$, but that the "distance" between $\hat{f}$ and $q(1 \mid x, a)$ is symmetric with respect to $A$; to see this, observe that it "compares" $\hat{f}$ to the Bayes optimal classifier for $q$, which itself may be skewed.[9]

### C.2   Key Proof Intuition for Theorem 4.2

One might try to obtain the PF by solving the constrained minimization $\hat{f}_{\underline{\Delta}} = \arg\min_{f \in \mathcal{F}} \mathcal{L}(p_{X,A,Y}, f)$ subject to $\Delta(p_{X,A,Y}, f) = \underline{\Delta}$, then computing $\mathcal{L}(p_{X,A,Y}, \hat{f}_{\underline{\Delta}})$ for all $\underline{\Delta}$. However, this problem is highly difficult to solve analytically and generally requires strong assumptions on the model class $\mathcal{F}$. Alternatively, one may wish to iterate over all possible $\hat{f}$ and compute their corresponding loss and fairness gap values (akin to [10, 24]), but this approach is only feasible for finite-dimensional, categorical features where the number of possible classifiers is small [40]; this approach does not scale for large models with high-dimensional, unstructured inputs.

To address these issues, we use a technique that often motivates in-processing methods and fair representation learning [35, 21, 15]: that one can induce a fairness gap $\underline{\Delta}$ by constructing an *artificial train distribution* $q \neq p$. In other words, we can transform the constrained minimization problem $\min_{f \in \mathcal{F}} \mathcal{L}(p_{X,A,Y}, f)$ subject to $\Delta(p_{X,A,Y}, f) = \underline{\Delta}$ into an unconstrained problem $\min_{f \in \mathcal{F}} \mathcal{L}(q_{X,A,Y}, f)$, where $q$ "tilts" $p$ so that the solution $\hat{f}$ to the latter problem has the desired fairness gap $\Delta(p_{X,A,Y}, \hat{f}) = \underline{\Delta}$. Note that "tilting" $q$ does *not* have any bearing on what distribution the developer actually uses to train their model—$q$ is an abstraction.[10] To provide further intuition,

---

[9]The second condition can be loosened to be equality with a constant shift, but we remove this for simplicity. It will introduce additive constants throughout our proof that are ultimately incorporated into the constants and thus do not affect the main result.

[10]We further emphasize that, although this discussion seems to suggest that we are restricting the allowable train distribution, that is not the case. We are simply imposing an *effective* train distribution as a proof technique to analytically characterize the PF.

recall that there is a known equivalence between minimizing an objective function subject to a constraint and minimizing the Lagrangian. Our approach leverages another notion of duality where minimizing $\mathbb{E}_p[\text{loss}]$ subject to a fairness constraint is equivalent to minimizing $\mathbb{E}_q[\text{loss}]$ for some $q$.

This change of measure allows us to characterize a Pareto-optimal classifer $\hat{f}$ for a given fairness gap $\underline{\Delta}$ in terms of the Bayes optimal classifer $q(1 \mid x, a)$ for the $q$ used to induce $\underline{\Delta}$. Then, we apply Theorem G.1, where Assumption 4.1 maps to the condition on $S$ in Theorem G.1. In summary, our approach alleviates the two challenges described above: for (i), we use the Bayes optimal classifier on the effective train distribution $q$ to simulate the constrained loss-minimization problem that is difficult to solve analytically; for (ii), we avoid imposing strong assumptions on the model class $\mathcal{F}$ by using a milder model symmetry assumption.

# D  Detailed Procedure for Applying the Scaling Law to the LDA Requirement

Here we give details for the high-level procedure described in Section 4.2. Given training data, test data, and a compute budget as well as an estimate of $N^+$ and $D^+$, as described in Section 3, a claimant would apply the scaling law as follows:

1. Given a small model class $N^-$ and small training dataset $D^-$, empirically trace out the loss-fairness PF for $N^-$ and $D^-$. To do so, train a set of models, each of which is obtained by minimizing the BCE loss on $\mathcal{D}^-$ plus a regularizer that encourages demographic parity. Varying the weight on the regularizer induces different fairness gaps. For each trained model, compute its loss and fairness gap on the given test data. Thus, each model marks a point on the loss vs. fairness gap plane. The lower convex hull of these points forms the empirical PF. The number of trained models as well as $N^-$ and $D^-$ should be chosen based on one's compute budget.

2. Repeat this for different values of $N^-$ and $D^-$, as permitted by one's compute budget.

3. Using these experiments, fit the constants $C_1$ through $C_7$ to each empirical PF using the following scaling law:

$$\begin{aligned}
\text{loss}(\underline{\Delta}) = {} & C_1 + C_2/(N^-)^{C_3} + C_2/(D^-)^{C_4} \\
& - C_5 \cdot C_6 \log(1 - C_7 + \underline{\Delta}) - C_5 \cdot (1 - C_6) \log(C_7 - \underline{\Delta}) \\
& + (C_7 - \underline{\Delta})(1 - C_7 + \underline{\Delta}) \left( \frac{C_5 \cdot C_6}{2(1 - C_7 + \underline{\Delta})^2} + \frac{C_5 \cdot (1 - C_6)}{2(C_7 - \underline{\Delta})^2} \right),
\end{aligned}$$

   with appropriate values for $N^-$ and $D^-$. Note that $C_3$ and $C_4$ are often set to $0.5$ [32, 7].

4. Once estimates of $C_1$ through $C_7$ have been obtained, one can extrapolate the PF for the contested model by using the same functional form as above, except substituting $N^+$ and $D^+$ for $N^-$ and $D^-$.

5. Alternatively, if one is given a specific loss-fairness pair $(\text{loss}(\underline{\Delta}), \underline{\Delta})$, one can use the functional form above to determine the values of $N^+$ and $D^+$ such that $(\text{loss}(\underline{\Delta}), \underline{\Delta})$ lies on the PF. This would tell the claimant how many resources the defendant needs to achieve that pair of values.[11]

Given that there are only 7 constants to fit, one could fit the scaling law by training as few as 7 small, high-quality models. In practice, the more models, the better (see discussion of Step 1 below).

**Note on implementation.**  We note that our approach "fails gracefully" in that it is *conservative*: if a claimant finds the contested model is $\delta$-far from the PF, then it is *at least* that far. There are two reasons why our approach is conservative. First, the empirical PF is always conservative by definition; there may be (and almost always are) models that Pareto dominate the models one finds empirically. Second, the expression in Theorem 4.2 is an inequality; it gives an upper bound on the loss of a Pareto-optimal model for a given fairness gap.

---

[11]Returning to Section 3, this analysis relies on the mild assumption that the PF improves monotonically as $N$ and $D$ increase. By our result (Theorem 4.2), this holds true because $B(\mathcal{F}, D)$, which is the only term that depends on the model class and dataset size, decreases as $N$ and $D$ increase.

**Considerations for Step 1.** There are multiple ways to trace out the PF empirically. For example, what we describe above is known as linear scalarization. Many works explore other methods (cf. Section 6). There are two main ways that the choice of method affects the claimant's results: (1) some methods are better at finding *Pareto-optimal* models across a wide range of fairness gaps, and (2) some methods find them more *efficiently* without having to train many models. Thus, a good method will help the claimant get as close to the true PF as possible (which can only help the claimant increase the gap between the contested model and the PF and thus strengthen their case) while training as few models as possible (and thus using as few resources as possible). Our scaling law is plug-and-play: as the methods for finding Pareto-optimal models efficiently improve, so too does our procedure.

**Considerations for Step 2.** Although not strictly necessary, re-running Step 1 on different values of $N^-$ and $D^-$ will generally improve the estimates of $C_1$ through $C_7$. There is a trade-off here: for a fixed compute budget, one can either run Step 1 multiple times with different $(N^-, D^-)$ values, or run it once with a $N^-$ and $D^-$ that are as close to $N^+$ and $D^+$ as possible. One should plan accordingly based on one's compute budget.

**Considerations for Step 3.** One may find that there are multiple sets of constants that fit the empirical PF well, as we explore in Section 5 and Appendix J. One could choose the set of constants that minimize the distance between the empirical PF and the fitted PF. Unintuitively, this might not be the appropriate choice. Recalling that the PF marks the lowest loss for each fairness gap, no observed point can lie below the true PF. Thus, one may wish to use the PF that is as close to the empirical one as possible while lying entirely below it.

Finally, we note that letting $B$ scale with $N^{-\alpha} + D^{-\beta}$ is well supported by previous works on large models (language models, in particular). As discussed in Section 1, our work is intended for large models, as this is where the LDA requirement imposes the greatest burden. One may wish to adjust the form for $B$ as appropriate.

## E  Discussion on the Practicality of the LDA Requirement

Several scholars critically examine the efficacy of disparate impact doctrine and practical application of the LDA requirement. Many observe that identifying precise and legally sufficient LDAs is difficult for plaintiffs [1, 55], and others highlight the inconsistent (and sometimes deferential) application of the business necessity standard by courts that can undermine LDA-based claims [26]. Some factors even discourage employers from seeking less discriminatory alternatives, including the risk of reverse discrimination lawsuits and legal uncertainty following *Ricci v. DeStefano* [28, 29]. Together, these critiques reveal a gap between the aspirations of disparate impact theory and the practical barriers faced by litigants. Our work takes the following perspective: while the LDA requirement remains intact, claimants increasingly need methods to demonstrate the existence of feasible alternatives.

## F  Helpful Lemmas

**Lemma F.1.** *Let $\mu$ and $\nu$ be finite probability measures of the same mass on a space $\mathcal{Z}$ such that $\mu$ is absolutely continuous with respect to $\nu$, i.e., $\mu \ll \nu$. Let $t(z) := \frac{d\mu}{d\nu}(z)$ denote the corresponding Radon-Nikodým derivative and $R(z) := t(z) - 1$. Let $S : \mathcal{Z} \to \mathbb{R}$ be absolutely integrable with respect to $\mu$ and $\nu$. Then,*

$$\mathrm{Cov}_\nu(R, S) = 0 \iff \mathbb{E}_\nu[RS] = \mathbb{E}_\nu[R]\mathbb{E}_\nu[S] \iff \mathbb{E}_\mu[S] = \mathbb{E}_\nu[S].$$

*Proof.* First, since $\mu$ and $\nu$ are probability measures and $t$ is the Radon-Nikodým derivative,

$$\mathbb{E}_\nu[R] = \mathbb{E}_\nu[t - 1] = \mathbb{E}_\nu[t] - 1 = 0.$$

Furthermore, by the definition of $R$,

$$\mathbb{E}_\nu[RS] = \mathbb{E}_\nu[(t-1)S] = \mathbb{E}_\nu[tS] - \mathbb{E}_\nu[S] = \mathbb{E}_\mu[S] - \mathbb{E}_\nu[S].$$

Therefore,

$$\mathbb{E}_\mu[S] = \mathbb{E}_\nu[S] \iff \mathbb{E}_\nu[RS] = 0 = \mathbb{E}_\nu[R]\mathbb{E}_\nu[S],$$

where the last equality is a consequence of having established that $\mathbb{E}_\nu[R] = 0$. This completes the proof, with the first $\iff$ in the lemma statement following from the definition of covariance. $\square$

**Lemma F.2.** *Consider the setup in Section 3, notation in Section 4, and data generating process in Theorem 4.2. Then,*

$$\text{Cov}_{p_{X|A=1}}(p(1 \mid X, A = 1), q(1 \mid X, A = 1)) \geq 0.$$

*Proof.* The covariance is well-defined since $p(1 \mid X, 1)$ and $q(1 \mid X, 1)$ have finite second moments by definition of $p$ and $q$. To show that the expression is non-negative, we appeal to Chebyshev's Association Inequality (see, e.g., [13, Theorem 2.14]), which states that if $f$ and $h$ are real-valued functions that are monotonic in the same direction and $Z$ is a real-valued random variable, then $\mathbb{E}[f(Z)h(Z)] \geq \mathbb{E}[f(Z)]\mathbb{E}[h(Z)]$. Thus, $\text{Cov}(i(Z), j(Z)) = \mathbb{E}[i(Z)j(Z)] - \mathbb{E}[i(Z)]\mathbb{E}[j(Z)] \geq 0$.

To get the final result, we map $Z$, $f$, and $h$ to quantities in our setup. Let the expectations be taken over $p(X|A = 1)$, let $\phi(X) = p(1|X, 1) = \sigma(g(X) - \zeta^p)$, and let $\psi(X) = q(1|X, 1) = \sigma(g(X) - \zeta)$.

Next, set $U = g(X)$ and denote by $\mu = p_{X|A=1} \circ g^{-1}$ the pushforward measure of $p_{X|A=1}$ by $g$. Introduce the deterministic functions $f(u) = \sigma(u - \zeta^p)$ and $h(u) = \sigma(u - \zeta)$ for $u \in \mathbb{R}$. Now, since the standard logistic (sigmoid) function $\sigma$ is monotonically increasing in its argument, both $f$ and $h$ are hence nondecreasing real-valued functions. So, Chebyshev's Association Inequality applies and gives that $\text{Cov}_\mu(f(U), h(U)) \geq 0$.

Finally, because $\phi(X) = f(U)$ and $\psi(X) = h(U)$, we consequently have that

$$\text{Cov}_{p_{X|A=1}}(\phi(X), \psi(X)) = \text{Cov}_\mu(f(U), h(U)) \geq 0,$$

which proves the claim. $\square$

Note that the lemma above is where the inequality in Theorem 4.2 arises. It holds with equality when $g(X)$ is almost surely constant.

**Lemma F.3.** *Suppose $Z \in (0, 1)$ is a random variable. Then,*

$$\mathbb{E}[\log(1 - Z)] = \log(1 - \mathbb{E}[Z]) - \frac{\text{Var}(Z)}{2(1 - \mathbb{E}[Z])^2} + \mathcal{O}\left(\mathbb{E}\left[(Z - \mathbb{E}[Z])^3\right]\right)$$

*and*

$$\mathbb{E}[\log(Z)] = \log(\mathbb{E}[Z]) - \frac{\text{Var}(Z)}{2\mathbb{E}[Z]^2} + \mathcal{O}\left(\mathbb{E}\left[(Z - \mathbb{E}[Z])^3\right]\right)$$

*Proof.* Since the function $\log(1 - z), z \in (0, 1)$ is (at least) three times differentiable, we can compute a second order Taylor expansion about a value $\mu \in (0, 1)$:

$$\log(1 - z) = \log(1 - \mu) - \frac{z - \mu}{1 - \mu} - \frac{(z - \mu)^2}{2(1 - \mu)^2} - \frac{1}{6} \cdot \frac{2}{(1 - \xi)^3}(z - \mu)^3$$

$$= \log(1 - \mu) - \frac{z - \mu}{1 - \mu} - \frac{(z - \mu)^2}{2(1 - \mu)^2} - \frac{(z - \mu)^3}{3(1 - \xi)^3}$$

for some $\xi$ between $\mu$ and $z$. Then, plugging in the random variable $Z$ for $z$ and $\mathbb{E}[Z]$ for $\mu$, and taking an expectation, we have

$$\mathbb{E}[\log(1 - Z)] = \log(1 - \mathbb{E}[Z]) - \frac{\mathbb{E}[(Z - \mathbb{E}[Z])^2]}{2(1 - \mathbb{E}[Z])^2} - \frac{\mathbb{E}[(Z - \mathbb{E}[Z])^3]}{3(1 - \xi_1)^3}$$

$$= \log(1 - \mathbb{E}[Z]) - \frac{\text{Var}[Z]}{2(1 - \mathbb{E}[Z])^2} + \mathcal{O}\left(\mathbb{E}[(Z - \mathbb{E}[Z])^3]\right)$$

where the linear term disappears from the first equality because its expectation is zero. Additionally, all expectations are well-defined because bounded random variables have finite moments of all orders.

The expansion for $\mathbb{E}[\log(Z)]$ is done similarly. $\square$

**Lemma F.4.** *Let $Z \in [0, 1], \mu = \mathbb{E}[Z], \mu_0 \in \mathbb{R}$, and $\Gamma \in \mathbb{R}$ subject to $\mu = \mu_0 - \Gamma$. Then,*

$$\text{Var}[Z] \leq (\mu_0 - \Gamma)(1 - \mu_0 + \Gamma).$$

*Proof.* Since $Z \in [0,1]$, $Z^2 \leq Z$, $\mathbb{E}[Z^2] \leq \mathbb{E}[Z]$. This implies that

$$\text{Var}[Z] = \mathbb{E}[Z^2] - \mathbb{E}[Z]^2 \leq \mathbb{E}[Z](1 - \mathbb{E}[Z]) = \mu(1 - \mu).$$

Plugging in $\mu = \mu_0 - \Gamma$ gives the result. $\qquad\square$

## G   Intermediate result

We provide an intermediate result that characterizes BCE loss for any DGP. We present this intermediate result for two reasons. First, it lends intuition for how our main result is obtained. Second, it does not depend on a specific definition of fairness or the DGP; it can therefore be used as a building block for future works that examine different notions of fairness and DGPs.

We use the same notation as in Section 4.1 and repeat it here for convenience. Let $p$ and $q$ denote joint distributions over random variables $X$, $A$, and $Y$. Let $p(1 \mid x, a)$ and $q(1 \mid x, a)$ denote the Bayes optimal classifiers under $p$ and $q$, respectively. Let $\text{KL}(\cdot, \cdot)$ denote the Kullback-Liebler divergence. The result below simply decomposes the loss of a classifier $\hat{f}$ on a test distribution $p$ into three components. For reasons that will become clear in the following section, we state this result in terms of $p$ and an auxiliary distribution $q$.

**Lemma G.1.** *Consider a classifier $\hat{f} : \mathcal{X} \times \mathcal{A} \to [0,1]$. Consider arbitrary joint distributions $p$ and $q$ over $X$, $A$, and $Y$, where $p_{X,A,Y} \ll q_{X,A,Y}$. Let $S(x,a,y) := y \log(q(1 \mid x,a)/\hat{f}(x,a)) + (1 - y) \log((1 - q(1 \mid x,a))/(1 - \hat{f}(x,a)))$ be absolutely integrable. We assume $\mathbb{E}_{p_{X,A,Y}}[S(x,a,y)] = \mathbb{E}_{q_{X,A,Y}}[S(x,a,y)]$. Then,*

$$\begin{aligned}
\mathcal{L}(p_{X,A,Y}, \hat{f}) &= (\mathcal{L}(q_{XAY}, \hat{f}) - \mathcal{L}(q_{XAY}, q_{1|X,A})) \\
&\quad + \mathbb{E}_{x,a \sim p_{X,A}} \left[ KL\left( p_{Y|x,a} \,\big\|\, q_{Y|x,a} \right) \right] + \mathcal{L}(p_{X,A,Y}, p_{1|X,A}).
\end{aligned} \tag{4}$$

**Interpretation.**   This result decomposes the loss of an arbitrary classifier $\hat{f}$ on a test distribution $p$ into three components: (1) the misspecification loss due to the choice of model class $\mathcal{F}$; (2) the loss due to distribution shift between $p$ and $q$; and (3) the irreducible test loss given by the BCE loss of the Bayes optimal classifier $p_{1|X,A}$ on $p_{X,A,Y}$, where $p_{1|X,A}$ denotes the classifier that returns $p_{1|X,A}(x,a)$ on $(x,a)$. This result follows from a simple telescoping sum that utilizes the definitions of information theoretic terms and the stated condition involving $S$.

Why is this useful? Recall that our goal is to derive a scaling law of the loss-fairness PF. That is, our goal is to produce a closed-form expression of loss, which is the left-hand side of (4), in terms of the fairness gap of $\hat{f}$ on $p$. Theorem G.1 gets us part of the way there; the next step is to write the right-hand side of (4) in terms of the chosen fairness gap for a given DGP.

**Understanding the condition on $\hat{f}$.**   We briefly comment on the condition $\mathbb{E}_{p_{X,A,Y}}[S(x,a,y)] = \mathbb{E}_{q_{X,A,Y}}[S(x,a,y)]$, which is a condition on $\hat{f}$. We observe that $S$ is the log likelihood ratio between Bernoulli models $q(1 \mid x,a)$ and $\hat{f}(x,a)$, such that $\mathbb{E}_{q_{X,A,Y}}[S(x,a,y)]$ is the expected conditional KL divergence between them. In the next section, Assumption 4.1 implicitly asks that this condition hold across all $q$'s that we consider, which implies that $\mathbb{E}_{q_{X,A,Y}}[S(x,a,y)]$, i.e., that the "distance" (as given by an expected log likelihood ratio) between $\hat{f}$ and the Bayes optimal $q(1 \mid x,a)$ is constant.

When would we expect this to hold true and is it restrictive? One might expect this to hold true if $\hat{f}$ is the model that results from training on $q$. If $\hat{f}$ is the result of training on $q$, then the condition asks that the training procedure and model class result in an $\hat{f}$ that replicates the patterns in its train distribution equally well, regardless of train distribution. In practice, we do not expect this condition to hold exactly; the training procedure and model class may fit some $q$'s better than others, e.g., if there exists a $q$ that is trivial to learn. However, we believe this is a reasonable condition for large models trained on sufficiently sophisticated tasks because deep learning models are designed to serve as universal function approximators across different distributions.

Importantly, we emphasize that $q$ is *not* the true train distribution. As discussed in Section 4.1, $q$ is an artificial, auxiliary distribution that gives rise to our main proof technique.

# H Proof of Lemma G.1

Recall that we denote the binary cross-entropy (BCE) loss $\ell$ of a classifier $f$ with respect to a sample $(x, a, y)$ by

$$\ell(f, (x, a, y)) = -\left(y \log(f(x, a)) + (1 - y) \log(1 - f(x, a))\right).$$

Note that $f$ can be a soft classifier that returns values in $[0, 1]$. Further recall that we denote the expected BCE of $f$ with respect to a distribution $p_{X,A,Y}$ by

$$\mathcal{L}(p_{X,A,Y}, f) = -\mathbb{E}_{x,a,y \sim p_{X,A,Y}}[y \log(f(x, a)) + (1 - y) \log(1 - f(x, a))],$$

and this notation $\mathcal{L}(\cdot, \cdot)$ is used analogously for other distributions and classifiers.

*Proof.* We split the proof into four steps.

**Step 1: Decomposition.** We can decompose the BCE loss of a classifier $\hat{f} \in \mathcal{F}$ on a distribution $p_{X,A,Y}$ as follows

$$H(p_{X,A,Y}, \hat{f}) = \underbrace{(H(p_{X,A,Y}, \hat{f}) - H(q_{XAY}, \hat{f}))}_{\text{loss of } \hat{f} \text{ due to distribution shift}} \tag{5}$$

$$+ \underbrace{(H(q_{XAY}, \hat{f}) - H(q_{XAY}, q_{1|X,A}))}_{\text{train loss of } \hat{f} \text{ relative to Bayes optimal}} \tag{6}$$

$$+ \underbrace{(H(q_{XAY}, q_{1|X,A}) - H(p_{X,A,Y}, q_{1|X,A}))}_{\text{loss of Bayes optimal } q_{1|X,A} \text{ due to distribution shift}} \tag{7}$$

$$+ \underbrace{(H(p_{X,A,Y}, q_{1|X,A}) - H(p_{X,A,Y}, p_{1|X,A}))}_{\text{difference in test loss of Bayes optimal classifiers}} \tag{8}$$

$$+ \underbrace{H(p_{X,A,Y}, p_{1|X,A})}_{\text{irreducible test loss}}, \tag{9}$$

by simply adding and subtracting terms, where we slightly abuse notation to let $q_{1|X,A}$ denote a (soft) classifier where the prediction for $(x, a)$ is given by $q_{Y|X,A}(1|x, a)$.

As written below each term, (5) can be viewed as the loss of $\hat{f}$ due to distribution shift between the train distribution $q_{X,A,Y}$ and test distribution $p_{X,A,Y}$; (6) is the difference in loss between $\hat{f}$ on the train distribution and the Bayes optimal classifier (also relative to the train distribution), which can be viewed as the loss due to the choice of model family $\mathcal{F}$; (7) also captures loss due to distribution shift, but for the Bayes optimal classifier of $q_{1|X,A}$; (8) gives difference in test loss between the Bayes optimal classifier that is optimal with respect to the train distribution and one that is optimal with respect to the test distribution, which can be viewed in some sense as the irreducible generalization loss; and (9) gives the irreducible test loss of the Bayes optimal classifier $p_{1|X,A}$ (that is optimal with respect the test distribution).

**Step 2: Expanding (5) and (7).** Focusing on just two of the terms above, we have

$$(5) + (7) = -\mathbb{E}_{x,a,y \sim p_{X,A,Y}}[y \log \hat{f}(x, a) + (1 - y) \log(1 - \hat{f}(x, a))]$$

$$+ \mathbb{E}_{x,a,y \sim q_{X,A,Y}}[y \log \hat{f}(x, a) + (1 - y) \log(1 - \hat{f}(x, a))]$$

$$- \mathbb{E}_{x,a,y \sim q_{X,A,Y}}[y \log q_{Y|X,A}(1|x, a) + (1 - y) \log(1 - q_{Y|X,A}(1|x, a))]$$

$$+ \mathbb{E}_{x,a,y \sim p_{X,A,Y}}[y \log q_{Y|X,A}(1|x, a) + (1 - y) \log(1 - q_{Y|X,A}(1|x, a))]$$

$$= -\int_{x,a,y} (p_{X,A,Y}(x, a, y) - q_{X,A,Y}(x, a, y))$$

$$\left(y \log \hat{f}(x, a) + (1 - y) \log(1 - \hat{f}(x, a))\right) dx\, da\, dy$$

$$+ \int_{x,a,y} (p_{X,A,Y}(x, a, y) - q_{X,A,Y}(x, a, y))$$

$$\left( y \log q_{Y|X,A}(1|x,a) + (1-y)\log(1 - q_{Y|X,A}(1|x,a)) \right) dx\, da\, dy$$

$$= \int_{x,a,y} (p_{X,A,Y}(x,a,y) - q_{X,A,Y}(x,a,y))$$

$$\left( y \log\left( \frac{q_{Y|X,A}(1|x,a)}{\hat{f}(x,a)} \right) + (1-y)\log\left( \frac{1 - q_{Y|X,A}(1|x,a)}{1 - \hat{f}(x,a)} \right) \right) dx\, da\, dy$$

$$= \int_{x,a,y} q_{X,A,Y}(x,a,y)(w(x,a,y) - 1)$$

$$\left( y \log\left( \frac{q_{Y|X,A}(1|x,a)}{\hat{f}(x,a)} \right) + (1-y)\log\left( \frac{1 - q_{Y|X,A}(1|x,a)}{1 - \hat{f}(x,a)} \right) \right) dx\, da\, dy$$

$$= \mathbb{E}_{q_{X,A,Y}} \left[ (w(x,a,y) - 1)\left( y \log\left( \frac{q_{Y|X,A}(1 \mid x,a)}{\hat{f}(x,a)} \right) \right.\right.$$

$$\left.\left. + (1-y)\log\left( \frac{1 - q_{Y|X,A}(1 \mid x,a)}{1 - \hat{f}(x,a)} \right) \right) \right],$$

where $w(x,a,y) = p_{X,A,Y}(x,a,y)/q_{X,A,Y}(x,a,y)$. Let

$$R := w(x,a,y) - 1 \quad \text{and} \quad S := y \log\left( \frac{q_{Y|X,A}(1 \mid x,a)}{\hat{f}(x,a)} \right) + (1-y)\log\left( \frac{1 - q_{Y|X,A}(1 \mid x,a)}{1 - \hat{f}(x,a)} \right).$$

Then, by invoking Theorem F.1—which applies because $S$ is absolutely integrable under the condition given in the lemma statement and because $p_{X,A,Y} \ll q_{X,A,Y}$—we have that

$$\mathbb{E}_{q_{X,A,Y}}[RS] = \mathbb{E}_{q_{X,A,Y}}[R]\mathbb{E}_{q_{X,A,Y}}[S] \iff \mathbb{E}_{q_{X,A,Y}}[S] = \mathbb{E}_{p_{X,A,Y}}[S].$$

By the condition given in the lemma statement, $\mathbb{E}_{q_{X,A,Y}}[S] = \mathbb{E}_{p_{X,A,Y}}[S]$ and hence we obtain

$$(5) + (7) = \mathbb{E}_{q_{X,A,Y}}[RS] = \mathbb{E}_{q_{X,A,Y}}[R]\mathbb{E}_{q_{X,A,Y}}[S] = 0.$$

where the last equality is because $\mathbb{E}_{q_{X,A,Y}}[R] = 0$, as shown in the proof of Theorem F.1.

**Step 3: Expanding** (8). The term (8) can be written as

$$-\int_{x,a,y} p_{X,A,Y}(x,a,y)\left[ y \log \frac{q_{Y|X,A}(1|x,a)}{p_{Y|X,A}(1|x,a)} + (1-y)\log \frac{q_{Y|X,A}(0|x,a)}{p_{Y|X,A}(0|x,a)} \right] dx\, da\, dy.$$

Substituting for possible values of $y \in \{0,1\}$ gives

$$-\int_{x,a} p_{X,A}(x,a)p_{Y|X,A}(0|x,a)\left[ \log \frac{q_{Y|X,A}(0|x,a)}{p_{Y|X,A}(0|x,a)} \right] dx\, da$$

$$-\int_{x,a} p_{X,A}(x,a)p_{Y|X,A}(1|x,a)\left[ \log \frac{q_{Y|X,A}(1|x,a)}{p_{Y|X,A}(1|x,a)} \right] dx\, da$$

$$= -\int_{x,a} p_{X,A}(x,a)(1 - p_{Y|X,A}(1|x,a))\left[ \log \frac{q_{Y|X,A}(0|x,a)}{p_{Y|X,A}(0|x,a)} \right] dx\, da$$

$$-\int_{x,a} p_{X,A}(x,a)p_{Y|X,A}(1|x,a)\left[ \log \frac{q_{Y|X,A}(1|x,a)}{p_{Y|X,A}(1|x,a)} \right] dx\, da$$

$$= -\int_{x,a} p_{X,A}(x,a)\left[ (1 - p_{Y|X,A}(1|x,a))\left[ \log \frac{1 - q_{Y|X,A}(1|x,a)}{1 - p_{Y|X,A}(1|x,a)} \right] \right.$$

$$\left. + p_{Y|X,A}(1|x,a)\left[ \log \frac{q_{Y|X,A}(1|x,a)}{p_{Y|X,A}(1|x,a)} \right] \right]$$

$$= \mathbb{E}_{x,a\sim p_{X,A}} \left[ \text{KL}\left( p_{Y|X,A}(1 \mid x,a) \,\|\, q_{Y|X,A}(1 \mid x,a) \right) \right],$$

where we use $\text{KL}\left( r \,\|\, s \right)$ to denote the KL divergence between the Bernoulli($r$) and Bernoulli($s$) distributions.

**Step 4: Putting it together.** Combining Steps 1-3,

$$
\begin{aligned}
H(p_{X,A,Y}, \hat{f}) \\
&= (5) + (6) + (7) + (8) + (9) \\
&= (6) + (8) + (9) \\
&= H(q_{XAY}, \hat{f}) - H(q_{XAY}, q_{1|X,A}) \\
&\quad + \mathbb{E}_{x,a \sim p_{X,A}} \left[ \mathrm{KL} \left( p_{Y|x,a} \,\|\, q_{Y|x,a} \right) \right] + H(p_{X,A,Y}, p_{1|X,A}),
\end{aligned}
$$

which concludes the proof. $\qquad\square$

# I  Proof of Theorem 4.2

*Proof.* In this proof, we use Theorem G.1 to express the BCE loss in terms of the train distribution parameter $\zeta$, then likewise express the demographic parity gap in terms of $\zeta$. Combining these representations allows us to express the BCE loss directly in terms of the demographic parity gap. We note that, by definition of our data-generating process, $p_{X,A,Y} \ll q_{X,A,Y}^{\zeta}$ for all $\zeta$.

**Step 1: Simplifying BCE loss.** Recall from Theorem G.1 that

$$
\begin{aligned}
H(p_{X,A,Y}, \hat{f}^{\zeta}) &= (H(q_{XAY}^{\zeta}, \hat{f}^{\zeta}) - H(q_{XAY}^{\zeta}, q_{1|X,A}^{\zeta})) \\
&\quad + \mathbb{E}_{x,a \sim p_{X,A}} \left[ \mathrm{KL} \left( p_{Y|x,a} \,\|\, q_{Y|x,a}^{\zeta} \right) \right] + H(p_{X,A,Y}, p_{1|X,A}).
\end{aligned}
$$

We begin by characterizing each component of this expression.

First, by Assumption 4.1, $H(q_{XAY}^{\zeta}, \hat{f}^{\zeta}) - H(q_{XAY}^{\zeta}, q_{1|X,A}^{\zeta})$ can be written as a constant $c_1(\mathcal{F}, D)$ that depends only on the model class $\mathcal{F}$ and the amount of training data $D$.

Second, we note that $H(p_{X,A,Y}, p_{1|X,A})$ can also be treated as a constant $c_2(p)$ that depends only on the test distribution $p$ and not on the model class $\mathcal{F}$ or the train distribution $q^{\zeta}$. This gives

$$
H(p_{X,A,Y}, \hat{f}^{\zeta}) = c_1(\mathcal{F}, D) + \mathbb{E}_{x,a \sim p_{X,A}} \left[ \mathrm{KL} \left( p_{Y|x,a} \,\|\, q_{Y|x,a}^{\zeta} \right) \right] + c_2(p), \tag{10}
$$

where

$$
\begin{aligned}
c_1(\mathcal{F}, D) &:= H(q_{XAY}^{\zeta}, \hat{f}^{\zeta}) - H(q_{XAY}^{\zeta}, q_{1|X,A}^{\zeta}), \\
c_2(p) &:= H(p_{X,A,Y}, p_{1|X,A}).
\end{aligned}
$$

Thus, it remains only to characterize the expected KL divergence term and write it in terms of the demographic parity gap. That will be the goal of the following steps.

**Step 2: Rewriting the KL divergence term.**

$$
\begin{aligned}
&\mathbb{E}_{x,a \sim p_{X,A}} \left[ \mathrm{KL} \left( p_{Y|x,a} \,\|\, q_{Y|x,a}^{\zeta} \right) \right] \\
&= -\int_{x,a} p_{X,A}(x,a) \Bigg[ (1 - p_{Y|X,A}(1|x,a)) \log \frac{1 - q_{Y|X,A}^{\zeta}(1|x,a)}{1 - p_{Y|X,A}(1|x,a)} \\
&\qquad\qquad\qquad\qquad\qquad + p_{Y|X,A}(1|x,a) \log \frac{q_{Y|X,A}^{\zeta}(1|x,a)}{p_{Y|X,A}(1|x,a)} \Bigg] \\
&= -\mathbb{E}_{x,a \sim p_{X,A}} [ (1 - p_{Y|X,A}(1|x,a)) \log (1 - q_{Y|X,A}^{\zeta}(1|x,a)) \\
&\qquad\qquad\qquad\qquad + p_{Y|X,A}(1|x,a) \log q_{Y|X,A}^{\zeta}(1|x,a) ] + c_3(p), \tag{11}
\end{aligned}
$$

where

$$
c_3(p) := \int_{x,a} p_{X,A}(x,a) \Big[ (1 - p_{Y|X,A}(1|x,a)) \log(1 - p_{Y|X,A}(1|x,a))
$$

$$+ p_{Y|X,A}(1|x,a)\log(p_{Y|X,A}(1|x,a))\Big],$$

is a constant that depends only on the test distribution $p$. Recalling from our data generating process given in Section 4 that $p(A = 1) = \pi$, (11) becomes

$$\mathbb{E}_{x,a\sim p_{X,A}}\left[\mathrm{KL}\left(p_{Y|x,a}\,\middle\|\,q^{\zeta}_{Y|x,a}\right)\right]$$
$$= -\mathbb{E}_{x,a\sim p_{X,A}}[(1 - p_{Y|X,A}(1|x,a))\log(1 - q^{\zeta}_{Y|X,A}(1|x,a))$$
$$+ p_{Y|X,A}(1|x,a)\log q^{\zeta}_{Y|X,A}(1|x,a)] + c_3(p)$$
$$= \pi\Big(-\mathbb{E}_{x\sim p_{X|A}}\Big[(1 - p_{Y|X,A}(1|x,A))\log(1 - q^{\zeta}_{Y|X,A}(1|x,A))$$
$$+ p_{Y|X,A}(1|x,A)\log q^{\zeta}_{Y|X,A}(1|x,A)\,\big|\,A = 1\Big]\Big) + c_4(p), \quad (12)$$

where the last line follows from the fact that, when $A = 0$, the $q$ distribution no longer depends on $\zeta$ (cf. our data generating process for $Y$ given in Section 4) such that

$$c_4(p) := c_3(p) + (1 - \pi)\Big(-\mathbb{E}_{x\sim p_{X|A}}\Big[(1 - p_{Y|X,A}(1|x,A))\log(1 - q^{\zeta}_{Y|X,A}(1|x,A))$$
$$+ p_{Y|X,A}(1|x,A)\log q^{\zeta}_{Y|X,A}(1|x,A)\,\big|\,A = 0\Big]\Big).$$

**Step 3: Upper bounding KL divergence term.** By the definition of covariance,

$$\mathbb{E}_{x\sim p_{X|A}}\Big[(1 - p_{Y|X,A}(1|x,A))\log(1 - q^{\zeta}_{Y|X,A}(1|x,A))\,\big|\,A = 1\Big]$$
$$= \mathbb{E}_{x\sim p_{X|A}}\Big[(1 - p_{Y|X,A}(1|x,A))\,\big|\,A = 1\Big]\mathbb{E}_{x\sim p_{X|A}}\Big[\log(1 - q^{\zeta}_{Y|X,A}(1|x,A))\,\big|\,A = 1\Big]$$
$$+ \mathrm{Cov}_{x\sim p_{X|A}}\Big((1 - p_{Y|X,A}(1|x,A)),\log(1 - q^{\zeta}_{Y|X,A}(1|x,A))\,\big|\,A = 1\Big).$$

By Lemma F.2, and our data-generating process as given in the theorem statement and Section 4, the covariance term is non-negative. Therefore,

$$\mathbb{E}_{x\sim p_{X|A}}\Big[(1 - p_{Y|X,A}(1|x,A))\log(1 - q^{\zeta}_{Y|X,A}(1|x,A))\,\big|\,A = 1\Big]$$
$$\geq \mathbb{E}_{x\sim p_{X|A}}\Big[(1 - p_{Y|X,A}(1|x,A))\,\big|\,A = 1\Big]\mathbb{E}_{x\sim p_{X|A}}\Big[\log(1 - q^{\zeta}_{Y|X,A}(1|x,A))\,\big|\,A = 1\Big].$$
$$(13)$$

One can apply analogous reasoning to show, again, by Theorem F.2 that

$$\mathbb{E}_{x\sim p_{X|A}}\Big[p_{Y|X,A}(1|x,A)\log q^{\zeta}_{Y|X,A}(1|x,A)\,\big|\,A = 1\Big]$$
$$\geq \mathbb{E}_{x\sim p_{X|A}}\Big[p_{Y|X,A}(1|x,A)\,\big|\,A = 1\Big]\mathbb{E}_{x\sim p_{X|A}}\Big[\log q^{\zeta}_{Y|X,A}(1|x,A)\,\big|\,A = 1\Big].$$

(Note that the two applications of Theorem F.2 above are where the inequality in Theorem 4.2 arises. Therefore, if one wishes to remove the inequality or provide a high-probability version of our result that holds with equality, one should address Theorem F.2.) Therefore, from (12),

$$\mathbb{E}_{x,a\sim p_{X,A}}\left[\mathrm{KL}\left(p_{Y|x,a}\,\middle\|\,q^{\zeta}_{Y|x,a}\right)\right]$$
$$\leq -\pi\Big(\mathbb{E}_{x\sim p_{X|A}}\Big[(1 - p_{Y|X,A}(1|x,A))\,\big|\,A = 1\Big]\mathbb{E}_{x\sim p_{X|A}}\Big[\log(1 - q^{\zeta}_{Y|X,A}(1|x,A))\,\big|\,A = 1\Big]$$
$$+ \mathbb{E}_{x\sim p_{X|A}}\Big[p_{Y|X,A}(1|x,A)\,\big|\,A = 1\Big]\mathbb{E}_{x\sim p_{X|A}}\Big[\log q^{\zeta}_{Y|X,A}(1|x,A)\,\big|\,A = 1\Big]\Big) + c_4(p)$$
$$= -\pi(1 - c_5(p))\mathbb{E}_{x\sim p_{X|A}}\Big[\log(1 - q^{\zeta}_{Y|X,A}(1|x,A))\,\big|\,A = 1\Big]$$
$$- \pi c_5(p)\mathbb{E}_{x\sim p_{X|A}}\Big[\log q^{\zeta}_{Y|X,A}(1|x,A)\,\big|\,A = 1\Big] + c_4(p),$$

where $c_5(p) := \mathbb{E}_{x\sim p_{X|A}}[p_{Y|X,A}(1|x,A)\,|\,A = 1]$.

**Step 4: Moving the expectation inside the log.** Then, we apply Theorem F.3, using $q^\zeta_{Y|X,A}(1 \mid x, 1)$ as the $Z$ in the statement of the lemma, and noting that it is a random variable that takes values between 0 and 1, as required by the lemma. For ease of notation, let

$$
\begin{aligned}
\bar{q}_1 &:= \mathbb{E}_{x \sim p_{X|A}} \left[ q^\zeta_{Y|X,A}(1|x, A) \mid A = 1 \right], \\
\bar{q}_0 &:= \mathbb{E}_{x \sim p_{X|A}} \left[ q^\zeta_{Y|X,A}(1|x, A) \mid A = 0 \right], \\
V &:= \mathrm{Var}_{x \sim p_{X|A}} \left[ q^\zeta_{Y|X,A}(1|x, A) \mid A = 1 \right].
\end{aligned} \tag{14}
$$

Then, Theorem F.3 gives

$$
\begin{aligned}
&\mathbb{E}_{x,a \sim p_{X,A}} \left[ \mathrm{KL}\left( p_{Y|x,a} \,\|\, q^\zeta_{Y|x,a} \right) \right] \\
&\leq -\pi(1 - c_5(p)) \left( \log(1 - \bar{q}_1) - \frac{V}{2(1 - \bar{q}_1)^2} \right) - \pi c_5(p) \left( \log(\bar{q}_1) - \frac{V}{2\bar{q}_1^2} \right) \\
&\quad + \mathcal{O}\left( \mathbb{E}_{x \sim p_{X|A}} \left[ (q^\zeta_{Y|X,A}(1 \mid x, A) - \bar{q}_1)^3 \mid A = 1 \right] \right) + c_4(p).
\end{aligned} \tag{15}
$$

Now that the KL divergence term is upper bounded, we proceed to characterize the demographic parity gap, and then substitute it into the KL divergence term.

**Step 5: Characterizing demographic parity gap.** The definition of demographic parity gap is

$$
\Delta(p_{X,A,Y}, \hat{f}^\zeta) := \left| \mathbb{E}_{x \sim p_{X|A}}[\hat{f}^\zeta(x, A)|A = 1] - \mathbb{E}_{x \sim p_{X|A}}[\hat{f}^\zeta(x, A)|A = 0] \right|.
$$

We can rewrite it as

$$
\begin{aligned}
&\mathbb{E}_{x \sim p_{X|A}}[\hat{f}^\zeta(x, A)|A = 1] - \mathbb{E}_{x \sim p_{X|A}}[\hat{f}^\zeta(x, A)|A = 0] \\
&\quad = \mathbb{E}_{x \sim p_{X|A}}[q^\zeta_{Y|X,A}(1|x, A)|A = 1] - \mathbb{E}_{x \sim p_{X|A}}[q^\zeta_{Y|X,A}(1|x, A)|A = 0] \\
&\qquad + (\mathbb{E}_{x \sim p_{X|A}}[\hat{f}^\zeta(x, A)|A = 1] - \mathbb{E}_{x \sim p_{X|A}}[q^\zeta_{Y|X,A}(1|x, A)|A = 1]) \\
&\qquad + (\mathbb{E}_{x \sim p_{X|A}}[q^\zeta_{Y|X,A}(1|x, A)|A = 0] - \mathbb{E}_{x \sim p_{X|A}}[\hat{f}^\zeta(x, A)|A = 0]).
\end{aligned}
$$

$$
\tag{16}
$$
$$
\tag{17}
$$

By Assumption 4.1,

$$
\begin{aligned}
&\mathbb{E}_{x \sim p_{X|A}}[\hat{f}^\zeta(x, A)|A = 1] - \mathbb{E}_{x \sim p_{X|A}}[\hat{f}^\zeta(x, A)|A = 0] \\
&\quad = \mathbb{E}_{x \sim p_{X|A}}[q^\zeta_{Y|X,A}(1|x, A)|A = 1] - \mathbb{E}_{x \sim p_{X|A}}[q^\zeta_{Y|X,A}(1|x, A)|A = 0].
\end{aligned}
$$

By our data generating process given in Theorem 4.2, in which $A$ mediates the effect of $\zeta$ and larger values of $\zeta$ result in lower positive classification rates,

$$
\mathbb{E}_{x \sim p_{X|A}}[q^\zeta_{Y|X,A}(1|x, A)|A = 0] \geq \mathbb{E}_{x \sim p_{X|A}}[q^\zeta_{Y|X,A}(1|x, A)|A = 1].
$$

for $\zeta \geq 0$. Therefore,

$$
\Delta(p_{X,A,Y}, \hat{f}^\zeta) = \bar{q}_0 - \bar{q}_1. \tag{18}
$$

Note that if we relaxed the second condition in Assumption 4.1 to allow for an additive constant, this would allow us to complete our analysis but with more bookkeeping, as the expression above would have two cases to ensure $\Delta \geq 0$.

**Step 6: Combining and rewriting BCE loss in terms of $\Delta(p_{X,A,Y}, \hat{f}^\zeta)$.** We can now substitute this expression for $\Delta(p_{X,A,Y}, \hat{f}^\zeta)$ from (18) into the BCE loss. From (15), we have

$$
\begin{aligned}
&\mathbb{E}_{x,a \sim p_{X,A}} \left[ \mathrm{KL}\left( p_{Y|x,a} \,\|\, q^\zeta_{Y|x,a} \right) \right] \\
&\leq -\pi(1 - c_5(p)) \log(1 - \bar{q}_0 + \Delta(p_{X,A,Y}, \hat{f}^\zeta)) + \frac{\pi(1 - c_5(p))V}{2(1 - \bar{q}_0 + \Delta(p_{X,A,Y}, \hat{f}^\zeta))^2}
\end{aligned}
$$

$$-\pi c_5(p)\log(\bar{q}_0 - \Delta(p_{X,A,Y}, \hat{f}^\zeta)) + \frac{\pi c_5(p)V}{2(\bar{q}_0 - \Delta(p_{X,A,Y}, \hat{f}^\zeta))^2}$$

$$+ c_4(p) + \mathcal{O}\left(\mathbb{E}_{x\sim p_{X|A}}\left[(q_{Y|X,A}^\zeta(1\mid x,A) - \bar{q}_1)^3 \mid A=1\right]\right).$$

We use Theorem F.4 with $Z = q_{Y|X,A}^\zeta(1\mid x,A), \mu = \bar{q}_1, \mu_0 = \bar{q}_0, \Gamma = \Delta = \bar{q}_0 - \bar{q}_1$ by (18), and that the expectations in Theorem F.4 are taken with respect to $p_{X|A=1}$ to get

$$V \le (c_6(p) - \Delta)(1 - c_6(p) + \Delta(p_{X,A,Y}, \hat{f}^\zeta)),$$

where $c_6(p) = \bar{q}_0$. Therefore,

$$\mathbb{E}_{x,a\sim p_{X,A}}\left[\text{KL}\left(p_{Y|x,a}\,\middle\|\,q_{Y|x,a}^\zeta\right)\right]$$

$$\le -\pi(1 - c_5(p))\log(1 - c_6(p) + \Delta(p_{X,A,Y}, \hat{f}^\zeta))$$

$$+ \frac{\pi(1 - c_5(p))(c_6(p) - \Delta(p_{X,A,Y}, \hat{f}^\zeta))(1 - c_6(p) + \Delta(p_{X,A,Y}, \hat{f}^\zeta))}{2(1 - c_6(p) + \Delta(p_{X,A,Y}, \hat{f}^\zeta))^2}$$

$$- \pi c_5(p)\log(c_6(p) - \Delta(p_{X,A,Y}, \hat{f}^\zeta))$$

$$+ \frac{\pi c_5(p)(c_6(p) - \Delta(p_{X,A,Y}, \hat{f}^\zeta))(1 - c_6(p) + \Delta(p_{X,A,Y}, \hat{f}^\zeta))}{2(c_6(p) - \Delta(p_{X,A,Y}, \hat{f}^\zeta))^2}$$

$$+ c_4(p) + \mathcal{O}\left(\mathbb{E}_{x\sim p_{X|A}}\left[(q_{Y|X,A}^\zeta(1\mid x,A) - \bar{q}_1)^3 \mid A=1\right]\right), \tag{19}$$

where we recall all constants:

$$c_2(p) = H(p_{X,A,Y}, p_{1|X,A}),$$

$$c_3(p) = \mathbb{E}_{x,a\sim p_{X,A}}\Big[(1 - p_{Y|X,A}(1|x,a))\log(1 - p_{Y|X,A}(1|x,a))$$

$$+ p_{Y|X,A}(1|x,a)\log(p_{Y|X,A}(1|x,a))\Big],$$

$$c_4(p) = c_3(p) - (1 - \pi)\mathbb{E}_{x\sim p_{X|A}}\Big[(1 - p_{Y|X,A}(1|x,A))\log(1 - q_{Y|X,A}^\zeta(1|x,A))$$

$$+ p_{Y|X,A}(1|x,A)\log q_{Y|X,A}^\zeta(1|x,A)\,\big|\, A=0\Big],$$

$$c_5(p) = \mathbb{E}_{x\sim p_{X|A}}[p_{Y|X,A}(1\mid x,A)\mid A=1]$$

$$c_6(p) = \mathbb{E}_{x\sim p_{X|A}}\left[q_{Y|X,A}^\zeta(1\mid x,A)\mid A=0\right]$$

Putting it all together, combining (10) and (19),

$$H(p_{X,A,Y}, \hat{f}^\zeta) \le B(\mathcal{F}, D) - c\cdot c'\log(1 - c'' + \underline{\Delta}) - c\cdot(1 - c')\log(c'' - \underline{\Delta})$$

$$+ (c'' - \underline{\Delta})(1 - c'' + \underline{\Delta})\left(\frac{c\cdot c'}{2(1 - c'' + \underline{\Delta})^2} + \frac{c\cdot(1 - c')}{2(c'' - \underline{\Delta})^2}\right)$$

$$+ \mathcal{O}\left(\mathbb{E}_{x\sim p_{X|A}}\left[(q_{Y|X,A}^\zeta(1\mid x,A) - \bar{q}_1)^3 \mid A=1\right]\right),$$

where

$$B(\mathcal{F}, D) := c_1(\mathcal{F}, D) + c_2(p) + c_4(p)$$

$$= c_1(\mathcal{F}, D) + H(p_{X,A,Y}, p_{1|X,A})$$

$$- (1 - \pi)\mathbb{E}_{x\sim p_{X|A}}\Big[(1 - p_{Y|X,A}(1|x,A))\log(1 - q_{Y|X,A}^\zeta(1|x,A))$$

$$+ p_{Y|X,A}(1|x,A)\log q_{Y|X,A}^\zeta(1|x,A)\,\big|\, A=0\Big]$$

$$+ \mathbb{E}_{x,a\sim p_{X,A}}\Big[(1 - p_{Y|X,A}(1|x,a))\log(1 - p_{Y|X,A}(1|x,a))$$

$$+ p_{Y|X,A}(1|x,a)\log(p_{Y|X,A}(1|x,a))\Big]$$

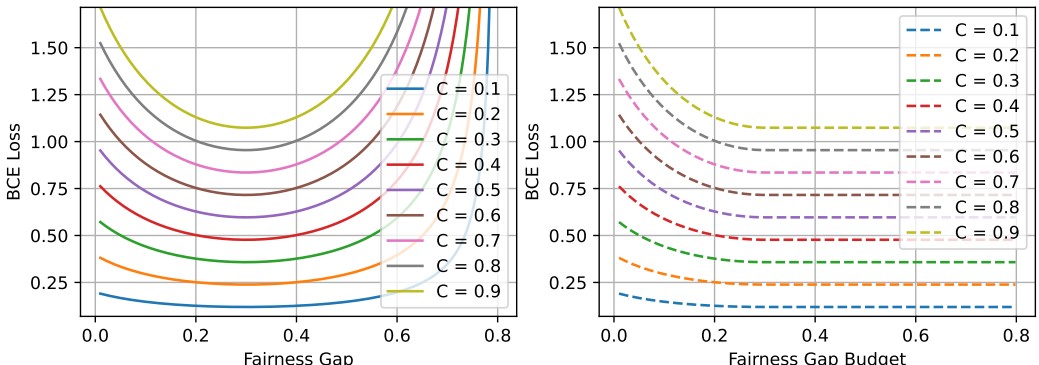

Figure 4: Simulations showing the shape of the closed-form expression for the Pareto frontier given in Theorem 4.2 for fixed values of $c' = 0.5$ and $c'' = 0.8$ while varying $c$. The left plot shows the precise closed-form, which is the lowest achievable loss among classifiers that have *exactly* the fairness gap value on the x-axis. The right plot shows lowest achievable loss among classifiers that satisfy the fairness gap *budget* on the x-axis.

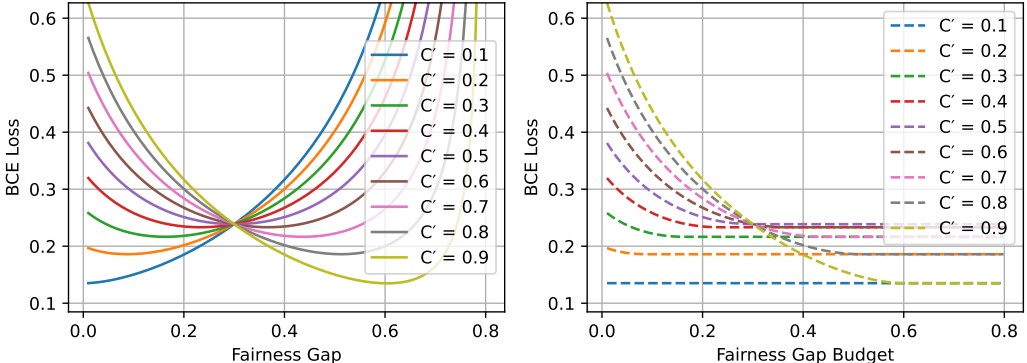

Figure 5: Simulations showing the shape of the closed-form expression for the Pareto frontier given in Theorem 4.2 for fixed values of $c = 0.2$ and $c'' = 0.8$ while varying $c'$. The left plot shows the precise closed-form, which is the lowest achievable loss among classifiers that have *exactly* the fairness gap value on the x-axis. The right plot shows lowest achievable loss among classifiers that satisfy the fairness gap *budget* on the x-axis.

$$c := \pi$$
$$c' := 1 - \mathbb{E}_{x \sim p_{X|A}}[p_{Y|X,A}(1 \mid x, A) \mid A = 1] \in [0,1]$$
$$c'' := \mathbb{E}_{x \sim p_{X|A}}[q^\zeta(1|x, A)|A = 0] \in [0,1]$$

$c, c', c'' \geq 0$. Note that the constants may not correspond exactly to the stated quantities above, e.g., due to the note below (18). $\qquad\square$

## J    Additional Experimental Details and Results

### J.1    Simulations

In this section, we provide further simulations.

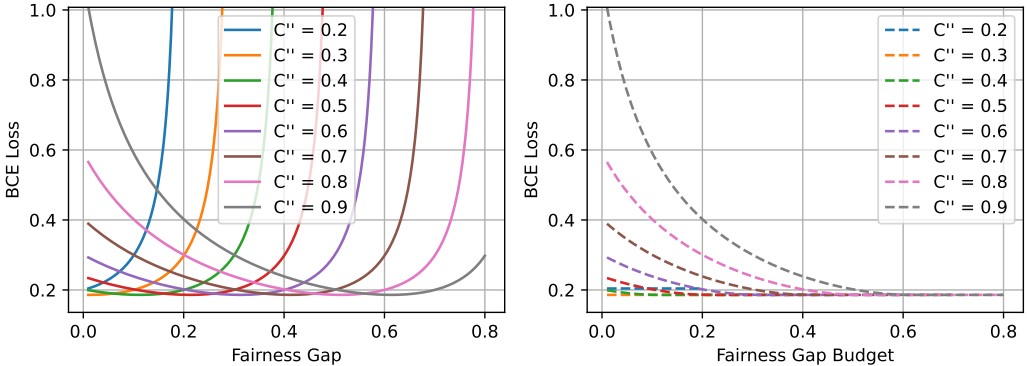

Figure 6: Simulations showing the shape of the closed-form expression for the Pareto frontier given in Theorem 4.2 for fixed values of $c = 0.2$ and $c' = 0.8$ while varying $c''$. The left plot shows the precise closed-form, which is the lowest achievable loss among classifiers that have *exactly* the fairness gap value on the x-axis. The right plot shows lowest achievable loss among classifiers that satisfy the fairness gap *budget* on the x-axis.

We begin by visualizing the behavior of the (3). Recall that (3) expresses loss as a function of the fairness gap $\underline{\Delta}$, where there are four constants: $B$, $c$, $c'$, and $c''$. In Figures 4 to 6, we show how the *shape* of (3) changes for different values of each constant, while keeping the other constants fixed and letting $B, \varepsilon = 0$. We provide further figures in Appendix J.

On the left, we plot the theoretically predicted Pareto frontier (PF) that is traced out by finding the lowest-loss classifier for an *exact* fairness gap $\underline{\Delta}$. Specifically, we evaluate the expression in Theorem 4.2, disregarding $\varepsilon$, over a range of $\underline{\Delta}$ values (typically from 0 to 1). As one might expect, the loss decreases as $\underline{\Delta}$ increases, then increases when a model is forced to have large fairness gap $\underline{\Delta}$.

To visualize the lowest loss achievable at or below a given fairness gap *budget* (rather than a fairness gap), we repeat the same simulation as in the left plot, then take the minimum loss value to the left of and including the current x-value (i.e., of the current fairness gap). Therefore, compared to the plot on the left, the plot on the right is, by definition, non-increasing. Intuitively, this gives the typical form of the PF by plotting the lowest achievable loss among classifiers with fairness gaps at or below the fairness gap budget given by the $x$-value.

We observe that increasing $c$ shifts the PF upwards while also increasing its curvature; changing $c'$ tilts the PF; and increasing $c''$ moves the PF to the right.

## J.2 Synthetic experiments

In this section, we provide additional details on the setup for the synthetic experiments in Figure 3 as well as additional results below.

## J.3 Model Architecture and Training

We used Pytorch to train our models. To test the scaling law, we trained models under 4 different MLP configurations, each with two hidden layers and ReLU activations, with a sigmoid output for binary classification:

- `[80, 80]` hidden units (8080 total parameters)
- `[160, 160]` hidden units (28960 total parameters)
- `[320, 320]` hidden units (109120 total parameters)
- `[640, 640]` hidden units (423040 total parameters)

For each architecture and $\lambda$ value, the data is split into training (65%), validation (15%), and test (20%) sets. The models are trained for 30 epochs using the Adam optimizer with a learning rate of

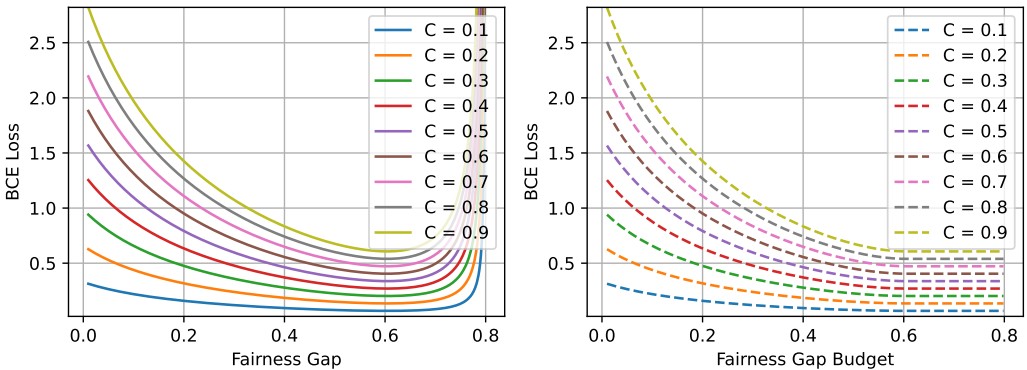

Figure 7: Simulations showing the shape of the closed-form expression for the Pareto frontier given in Theorem 4.2 for fixed values of $c'' = C_7 = 0.9$ and $c'' = 0.8$ while varying $c$. The left plot shows the precise closed-form, which is the lowest achievable loss among classifiers that have *exactly* the fairness gap value on the x-axis. The right plot shows lowest achievable loss among classifiers that satisfy the fairness gap *budget* on the x-axis.

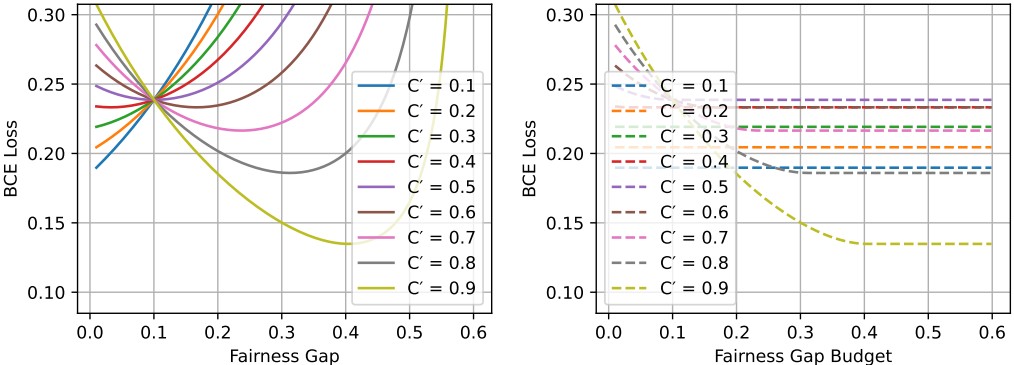

Figure 8: Simulations showing the shape of the closed-form expression for the Pareto frontier given in Theorem 4.2 for fixed values of $c = 0.2$ and $c'' = 0.6$ while varying $c'$. The left plot shows the precise closed-form, which is the lowest achievable loss among classifiers that have *exactly* the fairness gap value on the x-axis. The right plot shows lowest achievable loss among classifiers that satisfy the fairness gap *budget* on the x-axis.

0.001 and a batch size of 256, selecting the model with the lowest validation loss over the training epochs. We repeat each configuration over 3 random seeds and use all trials to find the Pareto frontier, resulting in 300 models per model size and 1200 models total.

### J.3.1 Additional Results

We test on two combinations of $\zeta$ and $\pi$ values.

We show results below. Interestingly, we found that the typical scaling law $N^{-0.5}$ did not necessarily fit our results perfectly, so we indicate the exponent used in each figure caption.

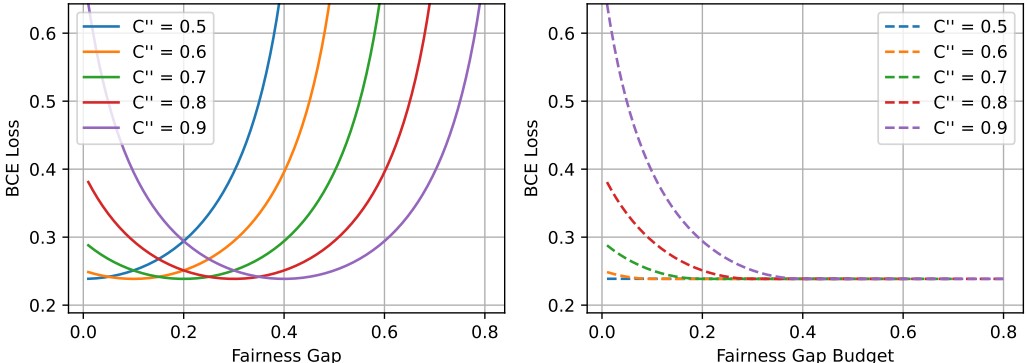

Figure 9: Simulations showing the shape of the closed-form expression for the Pareto frontier given in Theorem 4.2 for fixed values of $c = 0.2$ and $c'' = C_7 = 0.5$ while varying $c''$. The left plot shows the precise closed-form, which is the lowest achievable loss among classifiers that have *exactly* the fairness gap value on the x-axis. The right plot shows lowest achievable loss among classifiers that satisfy the fairness gap *budget* on the x-axis.

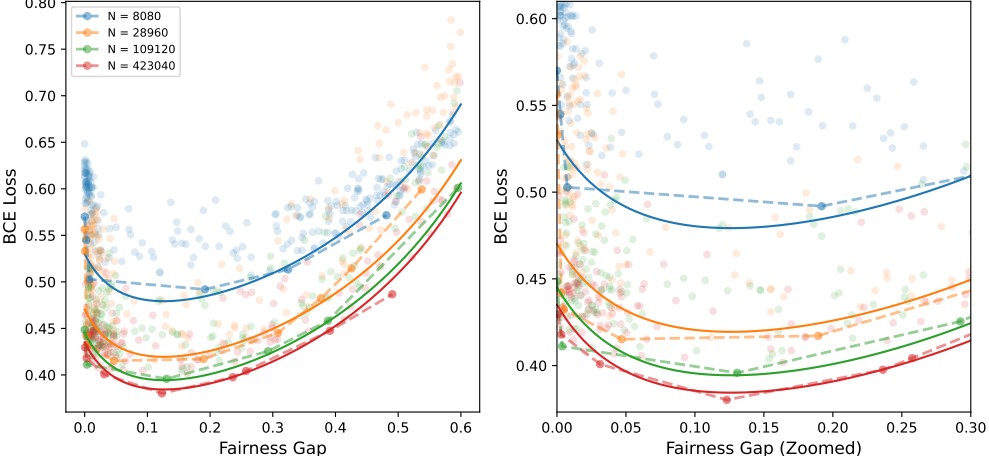

Figure 10: Pareto frontier for different model sizes under ($\pi = 0.2$) and bias strength $\zeta = 0.5$. Each point corresponds to a trained model with a different fairness regularization weight $\lambda$. The dashed lines show the empirical Pareto frontier, created by finding the lower convex hull of all the points. The solid lines show fitted curves to the points on the Pareto frontier using Theorem 4.2 with $c = C_5 = 0.16$, $c' = C_6 = 0.11$, $c'' = C_7 = 0.92$, with bias $C_1 = -0.285$, $C_2 = 55$, $C_3 = 0.7$, and $C_4 = 0.5$. The left panel shows the frontier across the range of $\Delta$, while the right panel zooms in on $\Delta \in [0, 0.3]$. The fitted curve mimics the empirical data well though it is imperfect. We found that there were many possible fits, depending on the precise choice. We show another possible fit in Figure 11 below.

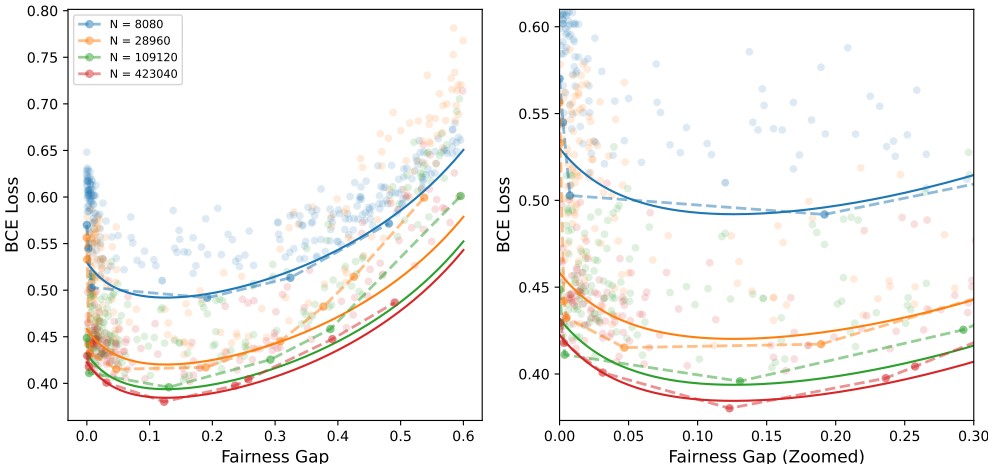

Figure 11: Pareto frontier for different model sizes under ($\pi = 0.2$) and bias strength $\zeta = 0.5$. Each point corresponds to a trained model with a different fairness regularization weight $\lambda$. The dashed lines show the empirical Pareto frontier, created by finding the lower convex hull of all the points. The solid lines show fitted curves to the points on the Pareto frontier using Theorem 4.2 with $c = C_5 = 0.12$, $c' = C_6 = 0.11$, $c'' = C_7 = 0.92$, with bias $C_1 = -1.205$, $C_2 = 150$, $C_3 = 0.8$, and $C_4 = 0.5$. The left panel shows the frontier across the range of $\Delta$, while the right panel zooms in on $\Delta \in [0, 0.3]$. The fitted curve is intentionally chosen to lower bound the larger models. That is, it fits to the small model curve, then uses it to extrapolate the larger model curve; we do so to exhibit one possible way to fit the curves since our training procedure was not optimized for the larger models and, as such, our empirical Pareto frontier is likely not optimal. We show another possible fit in Figure 3 below.

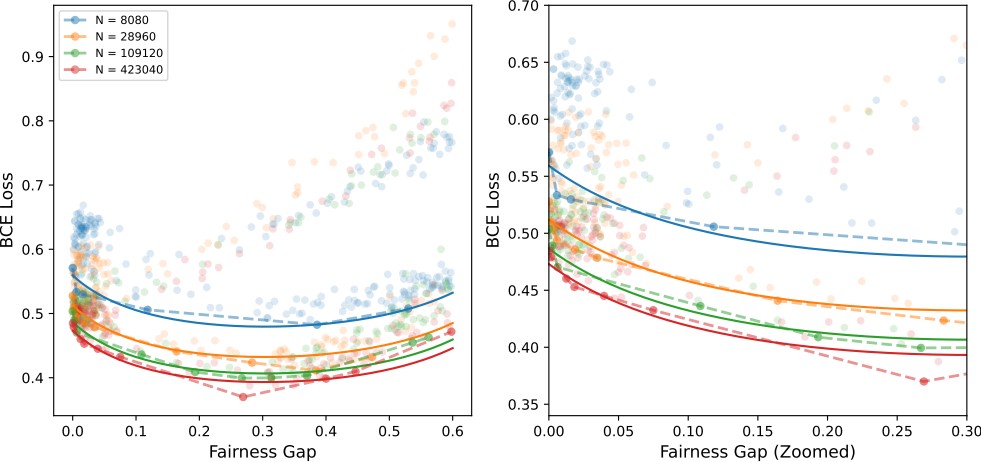

Figure 12: Pareto frontier for different model sizes under ($\pi = 0.4$) and bias strength $\zeta = 2$. Each point corresponds to a trained model with a different fairness regularization weight $\lambda$. The dashed lines show the empirical Pareto frontier, created by finding the lower convex hull of all the points. The solid lines show fitted curves to the points on the Pareto frontier using Theorem 4.2 with $c = C_5 = 0.08$, $c' = C_6 = 0.43$, $c'' = C_7 = 0.85$, with bias $C_1 = 0.195$, $C_2 = 9$, $C_3 = 0.5$, and $C_4 = 0.5$. The left panel shows the frontier across the range of $\Delta$, while the right panel zooms in on $\Delta \in [0, 0.3]$. The fitted curve mimics the empirical data well though it is imperfect. We found that there were many possible fits, depending on the precise choice. We show another possible fit in Figure 13 below.

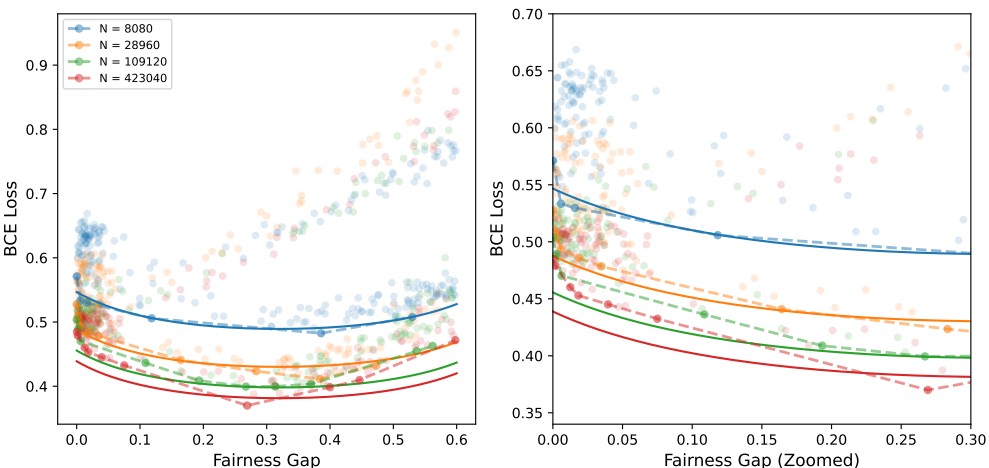

Figure 13: Pareto frontier for different model sizes under ($\pi = 0.4$) and bias strength $\zeta = 2$. Each point corresponds to a trained model with a different fairness regularization weight $\lambda$. The dashed lines show the empirical Pareto frontier, created by finding the lower convex hull of all the points. The solid lines show fitted curves to the points on the Pareto frontier using Theorem 4.2 with $c = C_5 = 0.06$, $c' = C_6 = C_7 = 0.5$, $c'' = C_7 = 0.82$, with bias $C_1 = 0.18$, $C_2 = 11.25$, $C_3 = 0.8$, and $C_4 = 0.5$. The left panel shows the frontier across the range of $\Delta$, while the right panel zooms in on $\Delta \in [0, 0.3]$. The fitted curve is intentionally chosen to lower bound the larger models. That is, it fits to the small model curve, then uses it to extrapolate the larger model curve; we do so to exhibit one possible way to fit the curves since our training procedure was not optimized for the larger models and, as such, our empirical Pareto frontier is likely not optimal. We show another possible fit in Figure 12.

