# OpenReview forum: "Audits Under Resource, Data, and Access Constraints: Scaling Laws For Less Discriminatory Alternatives"
_NeurIPS.cc/2025/Conference — NeurIPS 2025 poster_

### Official Review · Reviewer_4wHh · 2025-06-30

**Clarity:** 3
**Significance:** 3
**Originality:** 4
**Rating:** 5
**Confidence:** 3

**Summary:**

The authors propose a method for meeting the burden of evidence in algorithmic discrimination claims through proving the existence of “less discriminatory alternatives” (LDA). Specifically, AI models can be costly to produce in both time and resources, so the authors propose a method for estimating the achievable loss and fairness of large models. They do so by deriving an upper bound for the loss of a model depending on its achieved fairness score and the dataset and model size combined with known loss scaling laws for neural networks. Their theoretical results hold under somewhat strong assumptions and only one loss / fairness metric setting to serve as a proof of concept for this framework. The authors additionally provide numerical simulations to illustrate the intuition of their main theorem as well as validate their assumptions on the constants in their bound.

**Questions:**

- Is it non-trivial to bound constant $B(p,\mathcal{F},\mathcal{D})$ in Theorem 4.2 to explicate how $c_1(\mathcal{F},\mathcal{D})$ depends on $|\mathcal{F}|$ and $|\mathcal{D}|$?
- Under what loss functions and hypothesis classes are the scaling laws used in Section 5 proven to hold?
- Are the assumptions on the symmetry of the data generating process reasonable given that defendants have an incentive to release portions of their dataset that would in fact skew learning outcomes to show low loss is only achievable with high fairness gaps?

**Ethical Concerns:**

["NO or VERY MINOR ethics concerns only"]

**Final Justification:**

I have increased my score for this paper as a result of the thorough and engaging discussion the authors had with all of the reviewers. It is clear to me that this paper (with the planned clarity updates) will be useful for both a technical and legal audience and could inspire future work in an important area. My confidence remains the same because I am less familiar with how planned changes should weigh in the acceptance decision, so I will leave this up to the AC, though I will note that I am satisfied with the level of detail the authors have provided for their planned changes.

**Limitations:**

Yes.

**Paper Formatting Concerns:**

No.

**Quality:**

3

**Strengths And Weaknesses:**

$\textit{Strengths}$: The context of Title VII provides strong motivation for their problem set up. The authors uniquely address resource asymmetry concerns between plaintiffs and defendants, and this approach appears to be novel given the authors covered the relevant related works / concepts I could think of (Rashomon sets and trade secrecy concerns). The authors state their assumptions clearly and offer intuition on the technical assumptions as well as practical areas where the results would reasonably hold (e.g., when defendants are using neural networks). Additionally, their simulations provide initial evidence that it is reasonable to estimate Pareto frontiers using their bound by estimating constants with only a few model samples as opposed to traditional empirical PF tracing with many model samples.

$\textit{Weaknesses}$: One weakness in this paper is the lack of specificity on the conditions (e.g., loss function and hypothesis class and how it changes from scaling) under which the scaling laws for the constant $B(p,\mathcal{F},\mathcal{D})$ hold since those scaling laws drive the tractability of estimating the Pareto frontier from a few samples. Additional simulated experiments under a range of architectures and dataset distributions would be useful to further support this claim.  Another weakness is that training a small neural network can still be non-trivial, so this framework still requires plaintiffs to be able to produce properly trained (but small) neural networks.

$\textit{Minor formatting concerns}$: Use of $\mathcal{H}$ for loss instead of $\mathcal{L}$ in the appendix was a bit confusing when tracking where assumption 4.2 was used in the proof.

---

> ### Author Rebuttal · Authors · 2025-07-31
>
> Thank you for your very constructive feedback. We appreciate the time you took to provide interesting questions. Below we respond point‑by‑point and indicate the revisions we will make should the paper be accepted.
>
> **Clarification on contributions.** Before we address your specific concerns, we noticed some themes in reviews, so we wanted to clarify our contribution as well as how it meaningfully adds to existing work.
>
> 1. **Novel theoretical result that does not exist in extensive literature on Pareto frontiers**.  Despite the extensive interest on Pareto frontiers and recent work on performance-fairness Pareto frontiers [1-3, 11], no prior works have closed-form expressions for Pareto frontiers (in complex settings beyond, e.g., low-dimensional, binary features). The existing literature on performance-fairness Pareto frontiers falls into two categories: (i) clever methods to find the frontier empirically via various multi-objective optimization techniques [4-9]; and (ii) analytical approaches in simple settings that pre-date large models [10-12]. This is (i) not feasible in our setting, as it would still require that claimants are able to train models at the same scale as large companies and thus would place prohibitive costs on claimants with limited data and computational resources; and (ii) also not applicable in our setting, as it cannot be applied to **large** models with flexible, high-dimensional inputs. Thus, even absent our setting (LDAs), our derivation of a closed-form tight bound is a novel result in the study of Pareto frontiers, multi-objective optimization, and fairness that we provided by combining insights from information theory and results on implicit regularization via data augmentation/manipulation.
> 2. **Addressing a persistent issue in AI accountability and audits that appears repeatedly**. The authors engage with and belong to both the ML and legal communities, and have engaged with stakeholders who seek to audit and/or make legal claims about AI systems. Across these, there is a persistent issue (as in our introduction) of an asymmetry between the resources, information, data, and access that claimants possess vs. those that defendants possess. This asymmetry is especially pronounced in disparate impact procedures; namely, step 3 involving LDAs. At the moment, no method exists that is able to address this issue at scale (for **large** models). So this setting is a strong, motivating case study and we provide the only known framework that addresses the above asymmetry.
> 3. **Future work.** We note that while our main result on a closed-form scaling law holds for our stated assumptions and fairness as demographic parity, it is an important first step given the state of current work, as described above. We therefore hope our work provides a roadmap for extensions and hope to extend these results to other notions of fairness and performance. As our main contributions are our proposed reframing of LDAs for large models and the novel theoretical result, we leave extensive experiments to future work.
>
> We hope that these points help to clarify our contributions, and we will do our best to make these contributions clear/better contextual our contributions in a revision if given the opportunity.
>
> Next, we highlight and respond to specific concerns that you raised:
>
> - **Lack of specificity on the conditions under which the scaling laws hold.** Thank you for this point. We agree that our results, as presented, *apply only for our stated loss (BCE) and fairness definition (demographic parity).* As for the hypothesis class, the only assumption we place on it are the symmetry assumptions stated in our work (for an explanation of this assumption, see our last bullet below). Although our contribution in this work does not expand to other losses/fairness definitions, we believe it is an important first step and we seek to be upfront with our setup (i.e., that our results do not extend beyond it for now, except for what we show in the simulations, where we stress test the assumptions). Even so, we agree with the reviewer that the fact that our result lacks full generality is a limitation (though *we do not believe that all losses and fairness definitions will result the same closed-form,* so one would need to derive independent results for each setting). We will make this limitation even clearer in a revision, as we plan to include an explicit discussion and limitations section.
> - **More extensive experiments.** We agree with the reviewer that our experiments are limited. We hope that the points at the top of this rebuttal address this in part. To add to it, we make the following clarifications. (1) Our main contribution is the reframing of the LDA problem in low-resource/information settings and ensuing theoretical scaling law. We hope that the reviewer recognizes this as an important contribution, as there previous works *entirely on the empirics* of scaling laws and *entirely on the theory* of scaling laws that precede ours [13-16]. (2) We are already taking steps to engage with employment agencies to apply this framework to a real dataset. We believe this warrants a standalone follow-up work. (3) We want to note that, even so, we believe our synthetic experiments are important in that the main “gap” in our theoretical result is whether the assumptions hold. So we stress-test the assumptions in Section 5 via the synthetic experiments that test whether the theoretical result holds under a different data generating process than assumed, whether implicit regularization using training data manipulation is a good approximation of actual training, and if Assumption 4.2 holds in practice. Although we discuss this in the Appendix (E.2), we understand this should be clearer and will update Section 5 to reflect this.
> - **Training small neural nets is non-trivial.** We agree that training even small neural nets is non-trivial, but we want to emphasize our contribution compared to the status quo. At the moment, we are in a more dire situation in which a plaintiff must provide enough evidence that an LDA exists but often the resources/data involved to train one are impossible for them to obtain. They must produce some results that are convincing enough (if it’s too hand-wavy, it’s often not accepted), and thus it is impossible to avoid placing any burden on them.
> - **Formatting concerns.** We used H for the loss in the appendix proof but L in the Theorem 4.2 statement. Thank you very much for pointing this out—we will fix it in the revision.
> - **Non-trivial to bound constant B in Theorem 4.2 to explicate how $c_1(\mathcal{F}, \mathcal{D})$ depends on $ | \mathcal{F}|$ and $| \mathcal{D}|$.** We are not sure we fully understood the reviewer’s question, but under existing scaling laws (that only discuss loss), this term often follows the form $1/N^\alpha + 1/D^\alpha$, where $N = | \mathcal{F}|$ and $D = | \mathcal{D}|$.
> - **Reasonability of symmetry assumptions given incentive of defendant to only release skewed portions of their training data.** This is an interesting point. We believe that this assumption is fairly mild as it implies that the model class $\mathcal{F}$ is not biased/favorable towards any particular distribution. Although this assumption may not hold exactly in practice, *it is reasonable for large deep learning models, because they are intended to be universal function approximators.* That is, as a model class, they are designed to be relatively setting agnostic and over-parameterized such that they could be symmetric wrt the DGP. As for defendants being incentivized to release skewed portions of their training data, this type of dishonest behavior is impossible to prevent but *typically countered by the fact that failing to follow the court’s orders would be flagrantly breaking the law and thus we generally presume that the defendant will not do this or otherwise be severely penalized.* Again, preventing dishonesty is impossible in practice, but we typically use a “stick” to disincentivize it (e.g., penalties for perjury).
>
> [1] Bertsimas et al. (2011). The price of fairness. Oper. Res.
>
> [2] Menon & Williamson (2018). The cost of fairness in binary classification. FAccT.
>
> [3] Kim et al. (2020). FACT: A diagnostic for group fairness trade-offs. ICML.
>
> [4] Navon et al. (2020). Learning the Pareto front with hypernetworks. arXiv.
>
> [5] Ruchte & Grabocka (2021). Scalable Pareto front approximation for deep multi-objective learning. ICDM.
>
> [6] Singh et al. (2021). A hybrid 2-stage neural optimization for Pareto front extraction. arXiv.
>
> [7] Rothblum & Yona (2021). Consider the alternatives: Navigating fairness-accuracy tradeoffs via disqualification. arXiv.
>
> [8] Kamani et al. (2021). Pareto efficient fairness in supervised learning: From extraction to tracing. arXiv.
>
> [9] Liu & Vicente (2022). Accuracy and fairness trade-offs in machine learning: A stochastic multi-objective approach. CMS.
>
> [10] Xu & Strohmer (2023). Fair Data Representation for Machine Learning at the Pareto Frontier. JMLR.
>
> [11] Liang et al. (2021). Algorithm design: A fairness-accuracy frontier. arXiv.
>
> [12] Gillis et al. (2024). Operationalizing the search for less discriminatory alternatives in fair lending. FAccT.
>
> [13] Hestness et al. (2017). Deep learning scaling is predictable, empirically. arXiv.
>
> [14] Kaplan et al. (2020). Scaling laws for neural language models. arXiv.
>
> [15] Hoffmann et al. (2022). Training compute-optimal large language models. arXiv.
>
> [16] Bahri et al. (2024). Explaining neural scaling laws. PNAS.

---

> > ### Comment · Reviewer_4wHh · 2025-08-04
> > **Rebuttal Response - Addition of a hypothetical plaintiff using Theorem 4.2?**
> >
> > Thank you to the authors for their detailed response to each of my concerns and their thorough engagement with all of the reviewers. I have a few responses to the rebuttals to my stated weaknesses, especially in connection to the other reviewers' discussions.
> >
> > - (lack of specificity on the conditions under which the scaling laws hold / more experiments): I appreciate the authors' response and clarifications. I agree that it is more important for the experiments to address the feasibility of the assumptions used to establish the theoretical result, and I appreciate the pointer to Appendix E (and I agree that it would be useful to include some of this discussion in Section 5). Additionally, after reviewing the authors' discussion with Reviewer BDah, I believe starting with demographic parity makes a lot of sense for this kind of analysis in the context of disparate impact.
> > - (some burden on the plaintiff / potential dishonesty of defendant): These are fair points, and I appreciate that this work, in my opinion, is a significant positive step in the right direction compared to the status quo.
> > - (bound on B): Thank you for clarifying this. I was originally concerned that B could scale, say, exponentially with N or D, but after re-reading, I see how, there are accepted scaling laws for the BCE loss only, so it is reasonable to assume they would hold here too.
> >
> > I still believe this paper is a useful contribution that provides some framework for how complex algorithms like deep learning models could potentially be challenged in disparate impact claims. I have some familiarity with (anecdotal) stories of legal challenges to algorithmic harms, and I can appreciate the difficulties of the current status quo in challenging the use of complex algorithms. However, I do also agree with the other reviewers that this paper would benefit from a re-drafting to give more clarity about the technical assumptions and intuition behind the constants needed to establish the main result to broaden the audience of this paper. I do believe such a clarity pass would be sufficient for the results of this paper to be significant as-is without additional experiments, and I appreciate the author's suggested revisions in response to Reviewer BDah as a start.
> >
> > I'd like a little more clarity on one such suggestion - the step-by-step procedure of how to fit and apply a scaling law. Could such an explanation be connected to a walk through of how a hypothetical plaintiff could compute a bound from Theorem 4.2 using such scaling laws and knowledge of the loss / fairness gap of a smaller trained model?

---

> ### Author Response · Authors · 2025-08-05
> **Author response**
>
> A sincere thanks for your response. We further respond below:
>
> 1. (First bullet) We appreciate your update. If we are interpreting your response correctly, the suggestion is to incorporate further discussion into Section 5. We agree with this and will do so in the revision if given the opportunity. Our plan is to (i) make Section 5's main points and contributions much clearer (e.g., by adding \paragraph environments) and (ii) add a step-by-step explanation of how to fit a scaling law, linking that explanation to our simulations.
>
> 2. (Second and third bullets) We're glad that we were able to clarify these points.
>
> 3. (Improving clarity) We understand the feedback to improve clarity and carry out our promised revisions in this and other responses to reviewers. We take these promises seriously, as our goal is for this framework to be usable and applicable.
>
> 4. (Step-by-step procedure) Yes, absolutely. As mentioned above, we plan to incorporate a walk through/step-by-step procedure of precisely how one would use Theorem 4.2's result. At the moment, this is implied/scattered, but our goal would be to write it clearly in one place (either at the end of the main results section or at the start of the simulations section, which we may rename "Demonstration of Scaling Law" --> we plan to play around with both to determine which is clearest).
>
> [Apologies for the deleted response above. It was a response to a different reviewer that was mistakenly sent in this thread.]

---

> > ### Comment · Reviewer_4wHh · 2025-08-07
> >
> > Thank you for your follow-up comments and I am satisfied with your plan to describe the step-by-step procedure of applying Theorem 4.2. All the best with this paper!

---

### Official Review · Reviewer_ZUjW · 2025-07-01

**Clarity:** 3
**Significance:** 2
**Originality:** 3
**Rating:** 4
**Confidence:** 3

**Summary:**

This work formulates an analytical solution to finding less discriminatory alternative models under some assumptions. The authors introduce the problem setting of LDAs and some legal challenges in discovering them, specifically in asymmetries of data or model information. The authors provide some basic simulation results.

**Questions:**

Could you please motivate the terms in Thm 4.2 and give further definitions to what they represent?

**Ethical Concerns:**

["NO or VERY MINOR ethics concerns only"]

**Final Justification:**

Please see comment.

I am happy to recommend this at a weak accept, at a lower confidence of 3. Conceptually I'm happy with the work, but the technical appendix are beyond me.

*This review form does not seem to allow changing the reviewer confidence(?). This is very peculiar as we'd expect confidence to (hopefully) increase over the rebuttal period.

**Limitations:**

The authors have a good discussion on their model limitations, and specifically the scope of their work r.e. an initial fairness definition. Other analyses would have differing fairness implications which are outside of the scope of this work (and are OK).

**Quality:**

2

**Strengths And Weaknesses:**

Strengths

1. The authors give good legal and ethical motivations for their work.

2. The presentation of the work is overall quite good and easy to follow. The methods are straightforward and the problem is scoped to appropriate fairness measures.


Weaknesses

1. Overall, the authors do not motivate the hardness of their work, computationally. The authors do demonstrate practical limitations in LDA model discovery, but why is the problem challenging, if for example, we can make a few model assumptions and obtain Thm 4.2? Is Assumption 4.1 and 4.2 reasonable to make? Does 4.1 hold outside of Bayes optimal classifiers?

2. The authors miss important details of the primary Thm 4.2. Where is the intuition and justification for the constant terms? These are varied in Figure 2 and there's no intuition what varying C' ought to represent (let alone, C, C'').

3. Similarly, I don't understand the primary take-aways from  Fig 3. The authors describe model size 'N.' In number of nodes? They seemingly don't specify. But furthermore, does varying N solve 'limited resources' (D is fixed) or limited information? (again D is fixed).

Therefore, the authors don't directly demonstrate that they mitigate the use cases in their motivation.

4. I'm not convinced that the authors formulation is entirely correct. Specifically, in which applications is the LDA a responsibility of the plaintiff? And, what applications will there be *any* information on the vendor's model or data in a reduced capacity?

More likely is that a regulator or court requests a due-diligence report from the vendor. In applications such as healthcare or finance, regulators can demand alternative model search and require reporting from the vendor. In this case, the authors problem disappears with a different human solution.

Minor: The authors main claims (L69) could be edited to more succinctly convey the novelty.

---

> ### Author Rebuttal · Authors · 2025-07-31
>
> We appreciate your critique of both our theoretical and empirical work and the time you took with our submission. Below we respond to all of your points and note any revisions we will make shall the paper be accepted.
>
> **Clarification on contributions.** Before we address your specific concerns, we noticed some themes in reviews, so we wanted to clarify our contribution as well as how it meaningfully adds to existing work.
>
> 1. **Novel theoretical result that does not exist in extensive literature on Pareto frontiers**.  Despite the extensive interest on Pareto frontiers and recent work on performance-fairness Pareto frontiers [1-3, 11], no prior works have closed-form expressions for Pareto frontiers (in complex settings beyond, e.g., low-dimensional, binary features). The existing literature on performance-fairness Pareto frontiers falls into two categories: (i) clever methods to find the frontier empirically via various multi-objective optimization techniques [4-9]; and (ii) analytical approaches in simple settings that pre-date large models [10-12]. This is (i) not feasible in our setting, as it would still require that claimants are able to train models at the same scale as large companies and thus would place prohibitive costs on claimants with limited data and computational resources; and (ii) also not applicable in our setting, as it cannot be applied to **large** models with flexible, high-dimensional inputs. Thus, even absent our setting (LDAs), our derivation of a closed-form tight bound is a novel result in the study of Pareto frontiers, multi-objective optimization, and fairness that we provided by combining insights from information theory and results on implicit regularization via data augmentation/manipulation.
> 2. **Addressing a persistent issue in AI accountability and audits that appears repeatedly**. The authors engage with and belong to both the ML and legal communities, and have engaged with stakeholders who seek to audit and/or make legal claims about AI systems. Across these, there is a persistent issue (as in our introduction) of an asymmetry between the resources, information, data, and access that claimants possess vs. those that defendants possess. This asymmetry is especially pronounced in disparate impact procedures; namely, step 3 involving LDAs. At the moment, no method exists that is able to address this issue at scale (for **large** models). So this setting is a strong, motivating case study and we provide the only known framework that addresses the above asymmetry.
> 3. **Future work.** We note that while our main result on a closed-form scaling law holds for our stated assumptions and fairness as demographic parity, it is an important first step given the state of current work, as described above. We therefore hope our work provides a roadmap for extensions and hope to extend these results to other notions of fairness and performance. As our main contributions are our proposed reframing of LDAs for large models and the novel theoretical result, we leave extensive experiments to future work.
>
> We hope that these points help to clarify our contributions, and we will do our best to make these contributions clear/better contextual our contributions in a revision if given the opportunity.
>
> Next, we highlight and respond to specific concerns that you raised:
>
> - **Motivating computational hardness of work.** *Theorem 4.2 is the main result of the paper.* There is no easy way to obtain the scaling law, and this is precisely our contribution. To contextualize this contribution, there have been foundational papers that focus entirely on empirical scaling laws or entirely on the theory of scaling laws, as we discuss in Section 6. With our method, we show that one can obtain the scaling law specified in Theorem 4.2 and hence have an easier solution to the problem of finding an LDA. The proof of Theorem 4.2 was quite involved and, to the best of our knowledge, there exists no other closed-form characterizing the performance-loss Pareto frontier, and the framework of establishing LDAs under low resources and information using scalings is entirely new.
> - **Reasonability of Assumptions 4.1 and 4.2.** Assumption 4.1 is reasonable when considering very large models. It means that the model class $\mathcal{F}$ is not biased/favorable towards any particular distribution. Although this assumption may not hold *exactly* in practice (as most assumptions do not), it is a reasonable condition for deep learning models, especially large ones, because (large) deep learning models are meant to be universal function approximators. That is, as a model class, they are designed to be relatively setting agnostic and over-parameterized such that they could be symmetric wrt the DGP. A similar thought process holds for Assumption 4.2.
> - **Assumption 4.1 holding outside of Bayes optimal classifiers.** Assumption 4.1 uses \hat{f} to denote *any* classifier, not just a Bayes optimal classifier. Bayes optimality is only used as a proof technique to trace the Pareto frontier (it is not an assumption).
> - **Details of Thm 4.2, intuition and justification for constant terms.** The setup for Theorem 4.2 is provided between Sections 3.1 and 4.1. Constants (that may depend on complicated quantities like $p$) are a standard tool used in, e.g., statistical learning theory (see, e.g., Prop 2.5 or Thm 2.6 of [17]). There are two main reasons for this. (1) the exact value of the constant is context-specific! Providing a specific value would not be meaningful. (2) One could potentially provide a specific form for each constant, but that would require significant assumptions on the specific context. With respect to the scaling law literature, constants appear naturally to show that the scaling law takes a particular form, where certain constants must be “fit” based on data (e.g., see Equation 4.1 of [14]). Lastly, we note that one can trace the derivation of our constants in the proof on page 28 of the supplementary material.
> - **Take-aways from Fig 3, specify N, what does varying N solve.** N is the number of parameters of the model. We defined N at the start of the section on line 262 but will define it again near Figure 3. The magnitude of N relates to the computational difficulty of training such a model, so smaller N corresponds to limited computational resources. This general approach in finding a form of the scaling law as a function of N is common in the scaling law literature (e.g., Kaplan et al. 2020, Sharma and Kaplan 2022, or Clark et al. 2022).
> - **Correctness of formulation, responsibility for LDA.** Establishing an LDA **is the responsibility of the plaintiff in all disparate impact claims** under Title VII of the Civil Rights Act (see Lines 49-50). We would like to emphasize that the authors have significant engagement with the the legal community, and failing to ground our work in real stakeholder issues would be antithetical to our goals. This work was produced by first identifying a concern that is repeatedly raised, then seeking to provide a framework that can genuinely help plaintiffs and other claimants. We further point the reviewer to Sections 1, 2, and 6 (in particular, the “three steps of disparate impact cases” from lines 104-112). **Section 6 identifies several related works that write about the fact that the burden of establishing an LDA falls on plaintiffs.**
> - **Reduced information on vendor’s model.** To contextualize the situation, the current debate on burden of proof issues for claimants is whether or not vendors should release their *entire trained models* and *training data* and whether, in the absence of this, plaintiffs have any mechanisms for relief. Thus, asking vendors to release the approximate N and D is a much lower ask that we expect companies to welcome.
> - **Regulator or court requesting report from vendor.** While this may be true in some cases, in court cases involving disparate impact claims under Title VII of the Civil Rights Act, the burden of LDA proof *falls squarely on the claimant.* This is explicitly detailed in lines 110-112 and is precisely what we address. There is some discussion about whether this burden of search for an LDA should be shifted onto model developers (see lines 348-353), but for now the burden is entirely on the claimant.
>
> [1] Bertsimas et al. (2011). The price of fairness. Oper. Res.
>
> [2] Menon & Williamson (2018). The cost of fairness in binary classification. FAccT.
>
> [3] Kim et al. (2020). FACT: A diagnostic for group fairness trade-offs. ICML.
>
> [4] Navon et al. (2020). Learning the Pareto front with hypernetworks. arXiv.
>
> [5] Ruchte & Grabocka (2021). Scalable Pareto front approximation for deep multi-objective learning. ICDM.
>
> [6] Singh et al. (2021). A hybrid 2-stage neural optimization for Pareto front extraction. arXiv.
>
> [7] Rothblum & Yona (2021). Consider the alternatives: Navigating fairness-accuracy tradeoffs via disqualification. arXiv.
>
> [8] Kamani et al. (2021). Pareto efficient fairness in supervised learning: From extraction to tracing. arXiv.
>
> [9] Liu & Vicente (2022). Accuracy and fairness trade-offs in machine learning: A stochastic multi-objective approach. CMS.
>
> [10] Xu & Strohmer (2023). Fair Data Representation for Machine Learning at the Pareto Frontier. JMLR.
>
> [11] Liang et al. (2021). Algorithm design: A fairness-accuracy frontier. arXiv.
>
> [12] Gillis et al. (2024). Operationalizing the search for less discriminatory alternatives in fair lending. FAccT.
>
> [13] Hestness et al. (2017). Deep learning scaling is predictable, empirically. arXiv.
>
> [14] Kaplan et al. (2020). Scaling laws for neural language models. arXiv.
>
> [15] Hoffmann et al. (2022). Training compute-optimal large language models. arXiv.
>
> [16] Bahri et al. (2024). Explaining neural scaling laws. PNAS.
>
> [17] Wainwright MJ (2019). Basic tail and concentration bounds. In *High-Dimensional Statistics: A Non-Asymptotic Viewpoint*, pp. 21–57. Cambridge Univ. Press.

---

> ### Author Response · Authors · 2025-08-01
> **Response to reviewer**
>
> Thank you - we appreciate your time and thoughtful feedback, and we will do our best to improve the writing around the technical contributions.

---

### Official Review · Reviewer_BDah · 2025-07-02

**Clarity:** 2
**Significance:** 3
**Originality:** 3
**Rating:** 4
**Confidence:** 3

**Summary:**

This paper presents a novel approach towards legal cases of algorithmic discrimination, notably, cases where there is a need to demonstrate a less discriminatory alternative, and where the burden of proof falls on an entity with limited resources. The paper does an excellent job of motivating the approach and placing it in a concrete legal-operational context. The approach itself is solid and well-reasoned. However, the theoretical results themselves lack much-needed grounding and the simulations provided little added value due to their synthetic nature. There is also a lack of engagement with real use cases and known mechanisms of bias, which leads to a very big gap: it is very hard to estimate in which cases this approach will be operational. I felt that due to this, the paper misses the mark where it could have provided tangible, real-world value. I therefore suggest the authors to ground their work better prior to publishing, to achieve maximal impact of their result.

**Questions:**

- Can you find a few (or at least one) use cases that you can demonstrate your approach is superior to other approaches for this task?

**Ethical Concerns:**

["NO or VERY MINOR ethics concerns only"]

**Final Justification:**

Following the extensive discussion below, I am changing from 3 to 4, as I believe most of my concerns are due to a presentation issue rather than core issues. I am hesitant to fully recommend the paper without seeing the revised version, so I lowered my confidence from 4 to 3.

**Limitations:**

While the limitations the authors describe are correct, I think the discussion of their implications should be expanded (even in an appendix, if you feel like there is not enough space in the main body of the paper).

**Paper Formatting Concerns:**

No formatting concerns.

**Quality:**

3

**Strengths And Weaknesses:**

Strengths:

- The premise of the paper is very important and urgently needed.
- The clear connection to an existing legal framework is appropriate and well done
- The paper is well written
- The idea is (to my knowledge) novel, interesting and well reasoned.

Weaknesses:
- The theoretical results are not grounded in are real use cases. The requirements and limitations are put in general terms that require the reader to do much of their own work to try and find a single use case that this will work for. As a minimum, the authors need to demonstrate a class of use cases that would clearly benifit and ground their requirements and limitations concretely.

- The choice of algorithmic parity is not explained and is an odd choice for this settings. I am aware this could be swapped, but it is a measure that will be inappropriate to use in most use cases. It can also give the appearance of improving fairness while making outcomes worse for the marginalised group.
For example, let's say we have a University admissions algorithm that is trained on past admissions decisions, and has very limited past acceptances from schools in group A. We rebalance the data so we get more school A acceptances, which maintains accuracy (with respect to past decisions), and as such, we have made improvements based on your method. However, our algorithm is bad at detecting students from group A who are likely to succeed, so we are accepting more of those students, but setting them up to fail.

- Many of the limitations/complications related to the approach are not discussed. For example, what is the bias that comes from using a measure that is not well calibrated as ground truth? (e.g., Zilka et al. @ FAccT23'). Or, what if there is a better model, but it relies on different input data that is easily collected?

- The simulations would be much more valuable if they used real or even semi-real data, which seems very feasible to me.

---

> ### Author Rebuttal · Authors · 2025-07-31
>
> We sincerely appreciate your careful reading and constructive feedback. Below we address each concern and outline the concrete revisions we will make if the paper is accepted.
>
> **Clarification on contributions.** Before we address your specific concerns, we noticed some themes in reviews, so we wanted to clarify our contribution as well as how it meaningfully adds to existing work.
>
> 1. **Novel theoretical result that does not exist in extensive literature on Pareto frontiers**.  Despite the extensive interest on Pareto frontiers and recent work on performance-fairness Pareto frontiers [1-3, 11], no prior works have closed-form expressions for Pareto frontiers (in complex settings beyond, e.g., low-dimensional, binary features). Our work thus takes a step to fill this gap. To explain further, the existing literature on performance-fairness Pareto frontiers falls into two categories: (i) clever methods to find the frontier empirically via various multi-objective optimization techniques [4-9]; and (ii) analytical approaches in simple settings that pre-date large models [10-12], e.g., binary feature vectors of a fixed, low dimension. As we explained in our work, this (i) is not feasible in our setting, as it would still require that claimants are able to train models at the same scale as large companies and thus would place prohibitive costs on claimants with limited data and computational resources; and (ii) is also not applicable in our setting, as it cannot be applied to **large** models with flexible, high-dimensional inputs. Thus, even absent our setting/motivation (LDAs), our derivation of a closed-form tight upper bound is a novel result in the study of Pareto frontiers, multi-objective optimization, and fairness that we were able to provide by combining insights from information theory and results on implicit regularization via data augmentation/manipulation.
> 2. **Addressing a persistent issue in AI accountability and audits that appears repeatedly**. The authors engage with and belong to both the ML and legal communities. The authors have also engaged with stakeholders who seek to audit and/or make legal claims about AI systems. Across these conversations, there is a persistent issue (as identified in our introduction) of an asymmetry between the resources, information, data, and access that claimants possess vs. those that defendants/auditees possess. This asymmetry is especially pronounced in disparate impact procedures; namely, step 3 involving LDAs. At the moment, no method exists that is able to address this issue at scale (for **large** models). So this setting is a strong, motivating case study and we provide the only known framework that addresses the above asymmetry.
> 3. **Future work.** We note that while our main result on a closed-form scaling law holds for our stated assumptions and fairness as demographic parity, it is an important first step given the state of current work, as described above. We therefore hope our work provides a roadmap for extensions and hope to extend these results to other notions of fairness and performance. As our main contributions are our proposed reframing of LDAs for large models and the novel theoretical result, we leave extensive experiments to future work.
>
> We hope that these points help to clarify our contributions, and we will do our best to make these contributions clear/better contextual our contributions in a revision if given the opportunity.
>
> We now respond to specific concerns that you raised:
>
> - **Real application for theoretical results.** The setting of this paper is grounded in the specific Title VII procedure that arises in disparate impact doctrine (lines 104-112) and thus that procedure is a self-evident example.
> - **Experiments using real data.** Thank you for raising this. We hope that the points at the top of this rebuttal address this in part. To add to it, we make the following clarifications. (1) Our main contribution is the reframing of the LDA problem in low-resource/information settings and ensuing theoretical scaling law. We hope that the reviewer recognizes this as an important contribution, as there previous works *entirely on the empirics* of scaling laws and *entirely on the theory* of scaling laws that precede ours [13-16]. (2) We agree with the reviewer that an extensive real-world study is important to viability, and we are already taking steps to engage with employment agencies to apply this framework to a real dataset. We believe this warrants a standalone follow-up work. (3) We want to note that, even so, we believe our synthetic experiments are important as the main “gap” in our theoretical result is whether the assumptions hold. So the synthetic experiments test the assumptions in Section 5 to see whether the theoretical result holds under a different data generating process than assumed, whether implicit regularization using training data manipulation is a good approximation of actual training, and if Assumption 4.2 holds in practice. Although we discuss this in the Appendix (E.2), we understand this should be clearer and will update Section 5 to reflect this.
> - **Choice of algorithmic parity.** We agree with the reviewer that there is no universally “right” fairness definition. We want to assure the reviewer that the authors have engaged with the fairness community and with stakeholders on how the law, courts, and other policymakers have interpreted/applied fairness definitions in practice. Given this, we offer two points. (1) In practice, simple definitions like demographic parity and equalized odds are often more accepted by courts than sophisticated ones. (2) As discussed above, our work provides the first closed-form expression characterizing performance-fairness Pareto frontiers despite a long history of related work—we believe this is a significant first step that provides a roadmap for future work. However, we do acknowledge that our result is for a specific setting (choice of fairness metric, assumptions, model class) and we hope that it will be generalized in future work. We understand your example and will suggest that practitioners make sure the bias metric is appropriate for their use case.
> - **Bias from using a measure not as well calibrated as ground truth.** This is indeed a persistent issue that spans many fields. We appreciate the pointer and will add it to a limitations section. However, we do not believe that this issue precludes our framework from being applied. The concern the reviewer raises is, in fact, a more fundamental issue that *arises anytime an outcome-based fairness definition is used and thus is not an issue specific to our work,* though we fully agree that it is an important issue and would be a compelling extension.
> - **Better model that relies on easily collected but different input data.** This is certainly a possibility, but our motivating scenario is a company using a proprietary large model (so the plaintiff does not have access to any training material), and the task of training a competing large model is infeasible for the plaintiff given their limited resources and common appeals to trade secrecy used by companies. *This is the motivation of our work, which we describe in depth in Sections 1, 2, and 6.*
> - **Use cases in which our approach is superior to other approaches to this task.** We point the reviewer to the discussion at the top of this rebuttal as well as to our discussion of related work in Sections 6 and 2. Mainly, all other approaches characterize Pareto frontiers empirically (not feasible for low-resourced plaintiffs) or in settings with simple input spaces (not applicable to the increasingly large modern models that are the target of our work).
> - **Limitations.** We appreciate this feedback and are happy to discuss our limitations in depth in a revision by specifically addressing that (i) our method works conditional on the validity of the performance metric (e.g., demographic parity) supplied by the claimant; (ii) it also assumes that the claimant has at an ability to train small models (like an MLP) on small datasets; and (iii) our specific theoretical result contains several assumptions that should be further explained. We want to note that we strive in the existing manuscript to state all assumptions, definitions, and score clearly, as we agree that hiding these can only hurt practitioners (which is against the spirit of this work: to provide claimants with a useful framework).
>
> [1] Bertsimas et al. (2011). The price of fairness. Oper. Res.
>
> [2] Menon & Williamson (2018). The cost of fairness in binary classification. FAccT.
>
> [3] Kim et al. (2020). FACT: A diagnostic for group fairness trade-offs. ICML.
>
> [4] Navon et al. (2020). Learning the Pareto front with hypernetworks. arXiv.
>
> [5] Ruchte & Grabocka (2021). Scalable Pareto front approximation for deep multi-objective learning. ICDM.
>
> [6] Singh et al. (2021). A hybrid 2-stage neural optimization for Pareto front extraction. arXiv.
>
> [7] Rothblum & Yona (2021). Consider the alternatives: Navigating fairness-accuracy tradeoffs via disqualification. arXiv.
>
> [8] Kamani et al. (2021). Pareto efficient fairness in supervised learning: From extraction to tracing. arXiv.
>
> [9] Liu & Vicente (2022). Accuracy and fairness trade-offs in machine learning: A stochastic multi-objective approach. CMS.
>
> [10] Xu & Strohmer (2023). Fair Data Representation for Machine Learning at the Pareto Frontier. JMLR.
>
> [11] Liang et al. (2021). Algorithm design: A fairness-accuracy frontier. arXiv.
>
> [12] Gillis et al. (2024). Operationalizing the search for less discriminatory alternatives in fair lending. FAccT.
>
> [13] Hestness et al. (2017). Deep learning scaling is predictable, empirically. arXiv.
>
> [14] Kaplan et al. (2020). Scaling laws for neural language models. arXiv.
>
> [15] Hoffmann et al. (2022). Training compute-optimal large language models. arXiv.
>
> [16] Bahri et al. (2024). Explaining neural scaling laws. PNAS.

---

> > ### Comment · Reviewer_BDah · 2025-08-01
> > **Rebuttal acknowledgment**
> >
> > Thank you for your response to the review.
> >
> > I understand you believe the paper is strong enough as is (or with minor modifications), with the load mostly carried by the theoretical results. However, I still think that in its current version, it under-delivers on its potential strong impact.
> >
> > 1. Real application -- you are quoting a legal framework, which is not a demonstration of the method for a real use case -- I hope you do not believe your framework is immediately or reasonably easily applicable to any Title VII procedure?
> >
> > I note that highlighting a realistic use case does not have to take up a significant chunk of the paper (see Hron et al. ICLR 2023 -- box 1 as an example)
> >
> > 3. Algorithmic parity -- I find it hard to believe the courts accept this metric, as it is incompatible with most discrimination legislation. If you want to make this point, you have to be more convincing than saying the authors had conversations.
> >
> > I maintain that the work has clear strengths, but also noticeable gaps to be as impactful as it can be.

---

> ### Author Response · Authors · 2025-08-01
> **Author response**
>
> We appreciate your quick and thoughtful response.
>
> **Response to Point 1.** Given the imperfect nature of communicating via OpenReview, we will do our best to address what we think the reviewer is saying here. If we understand correctly, we completely agree with the spirit of the reviewer's point. What is often frustratingly clear is that the path to addressing issues that arise in CS/law is incredibly fraught and requires the buy-in of multiple stakeholders whose reactions are varied. This is actually precisely the motivation of our work, so we completely understand (and *share*) the skepticism that the framework is immediately applicable. Our hope is that this (and future works) can contribute to a **larger body work that is essentially adding "options" to the mix so that claimants/factfinders are no longer constrained by not having choices/tools to pick from**. They can survey and debate a variety of options, then select the one that seems best for the situation.
>
> So, to be fully clear, we do not feel that this would be *immediately* applicable, and it requires building a body of work around this issue, which always begins with initial steps (and we believe that this paper is helping take those steps). In fact, our goals are indeed much more modest (than hoping this would be immediately applicable): given the strong resistance to granting access, information, and data around AI systems, our hope is that, if our method yields results that support the plaintiff's claim, this would just help plaintiff **justify the request for more access** that allows them to prove (or disprove) that claim conclusively.*
>
> --
>
> **Response to Point 3 (or 2?).** Unfortunately, this is true! Although highly debated and very much not universally agreed upon, there is a long-standing "rule" known as the Four-Fifths rule, which compares selection rates and has been used in Title VII disparate impact cases. Although this standard might change, that does not change the reality of the situation. If one wants more recent evidence that that selection rates are still widely used, one needs only look at NYC's Local Law 144, which came into effect in 2023. To make the connection clear, the absolute difference in selection rates = the definition of demographic parity.
>
> --
>
> **A realistic use case.** We are still confused by this point. The broader issue that we seek to address is that a lack of access to models, training procedures, data, and resources has prevented plaintiffs from winning their cases (or, even earlier, from gathering enough evidence to bring a case). Our realistic use case *is* embedding this in Title VII, and even more specifically, scoping ourselves specifically to (i) disparate impact claims and (ii) the LDA requirement within disparate impact cases. We would argue that our entire paper is meant to tackle a specific case study.
>
> From here, making it more realistic is indeed a goal of ours, but this step (to our understanding) would be to empirically verify our framework on real data. This is, as mentioned, a very good suggestion and something we absolutely want to do! However, that step not within scope of this paper, as doing so would require multiple pages to describe how we acquired the data, how we cleaned it, issues that we ran into (which is always the case with real data), in-depth methodology, in-depth results, and ablations. To be clear, we don't think that the reviewer's point is that we should do this; the paragraph above is probably more applicable to addressing the reviewer's concern, and we believe this concern arises from a misunderstanding.
>
> Compared to the cited paper, ours is very different. The cited paper puts forth a model (a game) of the setting, then uses that example to demonstrate how the model captures appropriate settings that fall within scope of the motivation. This is fully natural and conventional for this type of paper, but ours is very different. Our motivation is a legal setting (in fact, a specific requirement within this setting), and we expend more than a figure (multiple sections!) motivating it.

---

> > ### Comment · Reviewer_BDah · 2025-08-02
> > **Response**
> >
> > To be clear, the point of attaching that paper was to provide an example of how a paper that mostly presents theoretical results gave intuition to what applying it to a use case might look like.
> >
> > The relevance to the two points I made is that what I want to see is similar in spirit, if you like. For example, if you took a real case or a realistic case where person A wanted to make a claim against company B and use your framework, what might that look like?
> >
> > -- Let's say that it's an LLM-based CV screening algorithm. What else will we need to know about it? Which experiments will we need to conduct? What will constitute "proof" in this case? Is it even applicable for LLM-based models? Will demographic parity be the right metric to use in such cases? (I am pretty sure the answer to the latter is no)
> >
> > Reading the paper in its current form, I am left with too many questions about applicability that I would expect you to be able to answer without doing real-data collection (I agree that will be amazing, but not necessary to make this paper strong).
> >
> > I know you think this can be resolved in future work, but I firmly believe this is needed for others to be able to take your work forward.

---

> ### Author Response · Authors · 2025-08-03
> **Response to reviewer**
>
> Thanks for your response. We do greatly appreciate your feedback - as you can understand, OpenReview is wonderful for allowing interaction, but it still is an imperfect way to communicate, so we are doing our best to fully understand your concerns given our reading of it. We always aim to sincerely engage with your (and reviewers') concerns.
>
> We acknowledge that you are unlikely to engage further given that you have submitted your mandatory acknowledgment, but we wanted to respond below to engage with your feedback.
>
> --
>
> **Efforts to further understand your call for a realistic use case.**
>
> Your clarification is greatly appreciated because we think we finally understand the point of confusion from both ends. **We believe that it may stem from an misunderstanding of the LDA setting we are tackling and would greatly appreciate the opportunity to explain why.** In the LDA setting, we are firmly embedded in the disparate impact doctrine. In that doctrine, nothing much changes from the classical classification/algorithmic scoring setting, beyond the fact that the models used are larger.
>
> For example, consider an LLM used to process resumes. They are still used to output score(s) and/or a recommendation to hire/not hire. This would be the same type of signal considered in classical discrimination/fairness works. The difference is that these large models can now effectively process/interpret complex, semantic text, like resumes or application short answers.
>
> The reviewer's clarification indicates that perhaps the reviewer is viewing legal definitions of LLM discrimination as definitions that interpret the text that is produced, perhaps whether the text contains undertones of discrimination. However, this is the portion that is due to a misunderstanding of our paper. Such analyses would not fall into disparate impact doctrine - it would be considered as **disparate treatment**. This would be completely out-of-scope for our paper, as our paper looks at Title VII --> Disparate Impact --> LDAs. The LDA requirement does not have any meaning beyond the disparate impact doctrine. (We make a short note of this in Footnote 3.)
>
> We hope this also addresses the reviewer's repeated concern about demographic parity not being the right metric. As we wrote above, demographic parity is indeed still used for disparate impact. Although there are countless other definitions for fairness and discrimination that have emerged (from equalized odds to individual fairness to envy freeness in matching markets to multicalibration), the evidence, as we wrote above, points to the fact that the courts often still prefer demographic parity. They favor it over concepts like equalized odds often because one cannot access the "ground truth" necessary to determine false positive rates/false negative rates.
>
> We would like to mention that, though we understand your clarification, we believe that we did our best to interpret your initial feedback about providing "a realistic use case". We hope you understand that, given that our work is *precisely motivated by a very specific, real use case*, we were confused. We did our best to understand and interpret it.
>
> --
>
>
> **Revisions to the work**
>
> Relatedly, we greatly appreciate the reviewer's time to give feedback, and it helps us understand where we need to clarify our work and setting. We plan to edit our work to reflect your feedback, including: (i) the scope of our setting, perhaps incorporating Footnote 3 into the main text, (ii) a figure or short example describing how this would apply to, say, the use of LLMs in employment decisions, (iii) a step-by-step procedure of how to fit and apply a scaling law.

---

> > ### Comment · Reviewer_BDah · 2025-08-03
> > **Still here!**
> >
> > First, this is very helpful. But did you edit your comment? I got an email with a different comment that was also helpful, l but now I can't find it.
> >
> > In any case, it does seem that I might not have fully understood the legal settings here -- I am less familiar with US discrimination legislation -- because I was thinking about a disparate treatment setting. But given that the audience for this conference is predominantly technical and international, I think there needs to be more hand-holding around this in the paper.
> >
> > I think the suggestions you mentioned would really help here. In addition, can you point me in the direction of 2-3 real court cases in which you think this procedure would have been applicable?
> >
> > On a more technical note. In the now edited comment, you mentioned that "the defendant should provide minimal information about the "type" of model used, the approximate model size N_target, the approximate training data size D_target, and a small portion of anonymized training data D' (even 1% to 5% of the training data would do)". But what about a test set? Is that to be provided by the Plaintiff?

---

> > > ### Author Response · Authors · 2025-08-04
> > > **Thanks for following up!**
> > >
> > > Yes! We did edit the comment - when re-reading everything, we revised our understanding of your feedback, so we adjusted it (we'll paste relevant portions of our previous comment at the end of these comments).
> > >
> > > Below, we respond to each of your points across multiple comments due to character limits. We want to first summarize to say that this engagement has been very helpful and that we will do our best to address these concerns in a revision if given the opportunity by: (i) better scoping disparate treatment/disparate impact; (i) moving the background section (currently Section 6) to be after the introduction and to incorporate more of the discussion below into it; and adding a specific use case, where we enumerate the expected process, how it would apply to something like an LLM employment tool, and what it means to fit our scaling law.
> > >
> > > —
> > >
> > > (1)  **Disparate impact vs. disparate treatment.** I'm glad we figured out one of the reasons we weren't converging! Indeed, disparate treatment has recently become of more interest in the CS community since LLMs, but prior to that, there had been relatively *longstanding agreement that what would apply to computational systems would be disparate impact* due to the inability to infer the intent of a computational system. You can think of disparate impact as **outcome-focused** and disparate treatment as intent-focused. (Note that even with LLMs, there's debate on whether disparate treatment is appropriate since we have no guarantees around the faithfulness of LLM outputs/explanations/CoTs, and there's no clear equivalent of "perjury" punishments for LLMs, so for now, the community still defaults to disparate impact.)
> > >
> > > --
> > >
> > > (2) **Re: more hand-holding in this paper due to audience.** We can absolutely go through the paper with a fine-tooth comb to ensure it does a bit more hand-holding. We want to mention that a lot of the discussion on disparate impact vs. disparate treatment, notions of fairness in CS vs. in the law, etc. are well discussed in the literature dating back to 2016.${}^\dagger$ As such, what we plan to do is bolster our Background and Related Work (which is current Section 6) with more discussion and references; move that section to right after the introduction; and clarify at points in our text (Sections 1-3) why we scope our attention to demographic parity for our illustration with references to the appropriate cases (e.g., those below).
> > >
> > > ${}^\dagger$In fact, from a historical perspective, the field of algorithmic fairness actually traces its origins back to Title VII and disparate impact —> employment discrimination inspired much if not most of the early research in fairness and discrimination, which then took a form of its own within the CS community. Since the 2016 paper “Big Data’s Disparate Impact” by Barocas and Selbst, hundreds of papers emerged on disparate impact, as well as papers on the applicability of disparate treatment. In our paper, we scope our attention, as described in Sections 1, 2, and 6 to LDAs in particular, which only arise in disparate impact cases. We do our best to zoom in just on this one aspect of the much larger field of fairness and discrimination in ML & AI.
> > >
> > >
> > > [More responses in the next comment]

---

> > > > ### Author Response · Authors · 2025-08-04
> > > > **[Response cont'd]**
> > > >
> > > > --
> > > >
> > > > (3) **Real court cases.** I can do you one better! Here's a page (one of many) on the **Four-Fifths Rule**: https://www.law.cornell.edu/cfr/text/29/1607.4 This rule is is set out by the Equal Employment Opportunity Commission (EEOC) which is responsible for enforcing employment anti-discrimination law (i.e., exactly what we're considering). This page discusses how one assesses impact (which is the outcome-based lens that disparate impact doctrine takes), and you can Ctrl-F for "selection rates" to see how it related to demographic parity. As we wrote above, there is debate on the Four-Fifths Rule, whether it is appropriate for ML, whether selection rates/demographic parity is the right notion of discrimination, etc. However, this type of debate is expected but does not invalidate our work (for a bad analogy, there's debate around many components of the Affordable Care Act, but methods that try to improve the implementation of the ACA despite its imperfections remain important, as do critiques of it). Given its current use, we believe that putting forth a result that uses demographic parity is well grounded in the law, though as we mention in the paper (and previous responses) that we absolutely hope to follow up with future work on other fairness notions, other performance notions, etc.
> > > >
> > > > For cases, there are many, as disparate impact doctrine has been around a while. We refer the reviewer to the original case *Griggs v. Duke Power Co (1971)* that we reference in our background section, as well as *Dothard v. Rawlinson (1977)*, *Ricci v. DeStefano (2009)*, *Connecticut v. Teal (1982)*, *EEOC v. Dial Corp. (2006)*, and many more.
> > > >
> > > > To point to how standard it is to consider **disparate impact** and to quantify it using **selection/pass rates**, see a recent law that uses selection rates for assessing automated employment tools (including algorithmic/AI tools for employment decisions) is **NYC's Local Law 144.** This is a well discussed (and debated) recent law that came into effect in 2023, and it is also *outcome-focused* (as I mentioned above, there has broad consensus to focus on impact-driven interpretations of discrimination). The rules of that law are here: https://rules.cityofnewyork.us/wp-content/uploads/2023/04/DCWP-NOA-for-Use-of-Automated-Employment-Decisionmaking-Tools-2.pdf . You can see that it also uses selection rates (not equalized odds, calibration, individual fairness, or the other notions that are popular in CS).
> > > >
> > > > We include some of this relevant discussion in 324-337, Footnote 3, Section 2, and the start of the Appendix.
> > > >
> > > > --
> > > >
> > > > (4) **Re: “on a more technical note…”** Yes, that was a typo - we’re assuming they would give a small portion of the data, both train and test.
> > > >
> > > > —
> > > >
> > > > (5) **Meta-note on the audience.** We completely appreciate the point about our audience. Although we know that this is not exactly what the reviewer is saying, we believe that there is space for papers in NeurIPS that seriously engage with non-technical, real-world challenges (in our case, legal challenges). Our incorporation of the law is to provide a *grounded* scenario in which plaintiffs are faced with limited (basically no) options at this point in time (scenario = establishing an LDA), and to use this scenario to motivate *a new framework that casts the problem as one of extrapolating the Pareto frontier,* then *proving it is indeed possible to solve this problem with limited resources,* even if it’s an existence proof for loss and demographic parity. We believe this grounding strengthens the work, rather than weakens it. We take the point that we can better walk our readers through the law and will do our best to do so (see above for our concrete steps).
> > > >
> > > > We also want to respond to the point about our audience being international. Although we ground ourselves in a specific scenario to demonstrate that we are not tackling an arbitrary but non-existent issue, LDAs are often implicated in other laws. We have discussed this with lawyers in prior discussions, but LDAs are often used when there is a trade-off at stake. Instead of simply saying “any statistical discrimination above this threshold is illegal”, LDAs offer a more balanced perspective that requires weighing two considerations at the same time. As such, LDAs appear in multiple areas, including in lending and disability law. We discuss some of this in Section 6 and the first section of the appendix. Beyond this, we have been in conversation with EU lawyers about the equivalent notions in the EU, with much interest from them.

---

> > > > > ### Author Response · Authors · 2025-08-05
> > > > > **[Response cont'd x2]**
> > > > >
> > > > > We promised to re-paste part of our edited response from a few messages ago. Here it is:
> > > > >
> > > > > In our case, we imagine the push and pull (which is usually many steps) would involve something like:
> > > > >
> > > > > 1. Plaintiff requests access to the model M and training data D to be able to produce an LDA
> > > > > 2. Defendant denies requests to the model M and training data D on the grounds of this being proprietary information (potentially appealing to trade secrecy) and even that revealing the training data D violates user privacy.
> > > > > 3. If the plaintiff wishes to use our procedure, they would express that, in the absence of M and D, the defendant should provide minimal information about the "type" of model used, the approximate model size N_target, the approximate training data size D_target, and a small portion of anonymized training data D' (even 1% to 5% of the training data would do).
> > > > > 4. If the defendant and judge agree, then the plaintiff runs our procedure, which is ...
> > > > > 5. Plaintiff trains small models of varying size on different fractions of D' and use this to fit the scaling law using Theorem 4.2. (Although this is a well known procedure, perhaps we took this for granted, and we can absolutely state the procedure for fitting a scaling law, which is to fit the 5 constants $C, C', C'', C'''$ and $C''''$ in Theorem 4.2, where $\mathcal{B} = C''' + C''''(N^{-1/2} + D^{-1/2})$, which we discuss briefly in Sections 4.3 and 5 but acknowledge was not laid out in a clear, step-by-step way.)
> > > > > 6. Once the scaling law has been fit, then the plaintiff plugs in N_target and D_target, which immediately gives the extrapolated Pareto frontier.
> > > > >
> > > > > At this point, the procedure supports the plaintiff's claim or doesn't. As we mentioned in "Response to Point 1" above, we do not know whether this will be enough to win the case, and if it is not, we hope that it is enough to justify *further* access (given the barrier in Step 2 above)

---

> > > > > > ### Comment · Reviewer_BDah · 2025-08-05
> > > > > > **Lots to unpack here!**
> > > > > >
> > > > > > Thanks for the detailed comments. I'll respond to most below, but give a summary of where I stand as a reviewer first. This discussion has convinced me that the work is appropriate and that the authors are very knowledgeable about the settings and context of the application of the work. My hesitation is only because the paper itself did not, and although the revisions you suggest are good, I would have liked to see the revised paper and ideally have another round of feedback, but alas, that is not the process for this conference.
> > > > > >
> > > > > > My current leaning is that while I cannot strongly endorse that paper without seeing the revised version, I do not want to block it either. This is because my reservations are no longer from a 'gatekeeper' perspective, and more from genuinely wanting this paper to maximise its potential attention and impact.
> > > > > >
> > > > > > I would clarify that I don't doubt your ability or knowledge, but as you are very familiar with your context, it can be really difficult to see that paper through the eyes of those who aren't, especially when adhering to the page limit...
> > > > > >
> > > > > > ----
> > > > > >
> > > > > > Disparate treatment and impact -- this is a very interesting discussion that probably deserves (and likely has) its own paper.  The way I understand disparate treatment in an algorithmic context is not via intent (as this, as you say, is not possible) but through the outcomes of counterfactuals. The issue with demographic parity is that when it is *enforced* in an algorithmic system, it may inadvertently cause disparate treatment, unless the population distribution is itself non-biased. (e.g., 50-50 male and female candidates, equally qualified). As such, taking demographic parity as are measure must carry some assumptions about the distribution, which in itself would carry assumptions about the representativeness of the data given by the defendant?
> > > > > >
> > > > > > Hand holding -- when I mean here is more about the logical flow rather than details. The example itself can really help here. But in some ways, this is more about making sure that both people with little legal concepts and those with little technical background can gain value from reading the paper. I wish I knew how to articulate it better. But I think I would have used the following stracture: -- explaining the legal context in laymen terms (including the distinction between treatment and impact). Then laying out how this translate to a technical context (assumptions, requirements, process, etc.), then presenting your results, then traslating this back to the application/process context.
> > > > > >
> > > > > > Relatedly, currently, I think the expirments provided appear somewhat trivial, in that they don't add much to the take away of the understanding of the paper. If you are adding an example, I'll make the expirements match the example (sticking to syntetic or semi-syntetic data), making everything tie together better.
> > > > > >
> > > > > > Court cases -- I think stating one or two real, recent cases that the procedure could have been used in (clarifying how it would have been used) would bring home the point that this is applicable.
> > > > > >
> > > > > > Regarding NeurIPS -- I wholeheartedly agree that there is a place for papers that seriously engage with non-technical, real-world challenges -- in fact, I wish there were many many more of these. I was not trying to imply it is a bad fit form the conference prespective. I would say, from my experience I think papers like yours can get lost in the biggness of NeurIPS, but those considertations are yours and outside my role as a reviewer.
> > > > > >
> > > > > > EU law -- I think an appendix on applicabliity to EU law will be really interesting and valuable!
> > > > > >
> > > > > > Last, but not least -- If accpeted (which is up to the AC in the end), often camera ready changes tend to be minimal due to time pressures and the feeling that the paper is "out". This is your call, but I want to emphasize that I have seen papers go from reject / marginal accept to top 5% of papers and even award winning papers by just making these type of changes -- I believe it's really worth the invesment!

---

> > > > > > > ### Author Response · Authors · 2025-08-06
> > > > > > > **Thanks for your engagement**
> > > > > > >
> > > > > > > Thank you for your deep engagement and thoughtful feedback throughout this process. We appreciate your perspective that the suggestions are really intended to improve readability, understanding, and therefore impact. We share your perspective and will do our absolute best to incorporate all feedback and suggestions, as we've discussed in our thread and as has appeared in several of the other threads.
> > > > > > >
> > > > > > > (Re: demographic parity, in case you're still interested! We agree and it's why people often prefer equalized TPRs or equalized TNRs or equalized odds. In practice, demographic parity becomes the fallback due to an inability to get "ground-truth" outcomes due to omission/selection bias, e.g., one cannot necessarily assess if an applicant who is not hired would have done well. All to say, it remains a challenging issue that some have studied (e.g., "Human Decision and Machine Predictions" and some works by Sharad Goel if I am remembering correctly).

---

### Official Review · Reviewer_9GNb · 2025-07-03

**Clarity:** 2
**Significance:** 3
**Originality:** 3
**Rating:** 4
**Confidence:** 3

**Summary:**

The authors focus on Title VII of the US Civil Rights Act and discuss their work in the context of legislation that utilizes the "less discriminatory alternatives" (LDA) framework. Under this framework, an individual claiming discrimination must be able to demonstrate the existence of an LDA with comparable performance. The authors highlight that claimants often lack the expertise or resources to find an LDA, hindering the efficacy of the legislation in facilitating accountability. To address this issue, the authors frame the problem as showing that there exists a model closer to the Pareto frontier. To circumvent computing the precise Pareto frontier, the paper proposes a scaling law for loss-fairness Pareto frontiers (under mild assumptions).

**Questions:**

- Figure 1 might be more compelling if it were from real data?
- What does it mean for a model class $\mathcal{F}$ to be symmetric with respect to the DGP?
- Can the authors provide an intuitive description/explanation on the quantities of “constants” in Section 4? i.e., C, C’, B(\cdot)
- Line 305
> including that $N$ and $D$ only affect the PF additively

Can the authors claim this when the experiment only varies $N$?

**Ethical Concerns:**

["NO or VERY MINOR ethics concerns only"]

**Final Justification:**

The authors were actively engaged throughout the rebuttal process and have addressed most of my concerns involving clarity in theoretical results and empirical demonstrations. I am in favor of accepting the paper, provided that the authors take the necessary steps to revise the paper for their camera-ready as promised. I have changed the score to a "weak accept". But I have lowered my confidence since

1. I do not know how the authors will execute the revision; and
2. Not sure if it was rushed, but the tone (language) and format of the original submission was sub-par. I am not 100% confident that the authors can address this (they have avoided answering this in their response apart from changing Figure 1).

**Limitations:**

Yes

**Paper Formatting Concerns:**

Formatting issues are not major. Please see Strengths and Weaknesses

**Quality:**

2

**Strengths And Weaknesses:**

## Strengths
The authors have framed the problem very well. The problem they aim to address is significant and seems to be up to date with the literature.

## Weaknesses
**Structure and Tone**
The structure and tone of the work leave a lot to be desired for an academic paper. This is immediately apparent with Figure 1 (it should not be difficult to recreate this plot digitally) and the fact that related works are at the very end. Section 1 does not have any citations as well. The numbered points in the main contribution should be more concise. The tone of the paper is not consistently professional (i.e., line 127, line 260).

**Clarity in Theoretical Results**
Overall, the theoretical results in Section 4 are hard to follow because many variables are not clearly defined (see questions for specific examples). There are a lot of variables and custom notation (which is fine) that are mentioned in passing without elaboration on what they mean. This is crucial considering the paper aims to provide a method for those who may not be well-versed in machine learning. I would imagine that they would not have the easiest time understanding how they should adjust parameters in the framework.

**Leaving Empirical Validation for Future Work**
This is the main weakness of this paper. Considering how the authors have positioned this paper in specific legislation, I expected empirical results (from non-synthetic datasets). Without empirical results, the authors have put themselves at a disadvantage: it is difficult to argue for the utility of the paper's contribution. The lack of clarity in theoretical results could have been resolved through a guided demonstration on how one might use the proposed PF construction.

I suspect that this may be because the paper focuses on "large models" or "universal approximators." But I want to ask why the authors have focused on large models. Are they widely used in areas where LDA-esque legislation is in effect?

---

> ### Author Rebuttal · Authors · 2025-07-31
>
> Thank you for your thorough and constructive feedback. We appreciate the time you took to engage with our submission. Below we respond point‑by‑point and indicate the revisions we will make in the camera‑ready version should the paper be accepted.
>
> **Clarification on contributions.** Before we address your specific concerns, we noticed some themes in reviews, so we wanted to clarify our contribution as well as how it meaningfully adds to existing work.
>
> 1. **Novel theoretical result that does not exist in extensive literature on Pareto frontiers**.  Despite the extensive interest on Pareto frontiers and recent work on performance-fairness Pareto frontiers [1-3, 11], no prior works have closed-form expressions for Pareto frontiers (in complex settings beyond, e.g., low-dimensional, binary features). Our work thus takes a step to fill this gap. To explain further, the existing literature on performance-fairness Pareto frontiers falls into two categories: (i) clever methods to find the frontier empirically via various multi-objective optimization techniques [4-9]; and (ii) analytical approaches in simple settings that pre-date large models [10-12], e.g., binary feature vectors of a fixed, low dimension. As we explained in our work, this (i) is not feasible in our setting, as it would still require that claimants are able to train models at the same scale as large companies and thus would place prohibitive costs on claimants with limited data and computational resources; and (ii) is also not applicable in our setting, as it cannot be applied to **large** models with flexible, high-dimensional inputs. Thus, even absent our setting/motivation (LDAs), our derivation of a closed-form tight upper bound is a novel result in the study of Pareto frontiers, multi-objective optimization, and fairness that we were able to provide by combining insights from information theory and results on implicit regularization via data augmentation/manipulation.
> 2. **Addressing a persistent issue in AI accountability and audits that appears repeatedly**. The authors engage with and belong to both the ML and legal communities. The authors have also engaged with stakeholders who seek to audit and/or make legal claims about AI systems. Across these conversations, there is a persistent issue (as identified in our introduction) of an asymmetry between the resources, information, data, and access that claimants possess vs. those that defendants/auditees possess. This asymmetry is especially pronounced in disparate impact procedures; namely, step 3 involving LDAs. At the moment, no method exists that is able to address this issue at scale (for **large** models). So this setting is a strong, motivating case study and we provide the only known framework that addresses the above asymmetry.
> 3. **Future work.** We note that while our main result on a closed-form scaling law holds for our stated assumptions and fairness as demographic parity, it is an important first step given the state of current work, as described above. We therefore hope our work provides a roadmap for extensions and hope to extend these results to other notions of fairness and performance. As our main contributions are our proposed reframing of LDAs for large models and the novel theoretical result, we leave extensive experiments to future work.
>
> We hope that these points help to clarify our contributions, and we will do our best to make these contributions clear/better contextual our contributions in a revision if given the opportunity.
>
> Next, we highlight and respond to specific concerns that you raised:
>
> - **Re: Figure 1****,** we thank the reviewer for pointing this out and will replace it with a more professionally rendered schematic. Regarding the suggestion to use empirical data, our objective with Figure 1 is to provide a schematic illustrating *why the legal problem finding LDAs can be cast/formalized in terms of Pareto frontiers* and *the intuition behind scaling laws for Pareto frontiers.* This aims to provide the reader with a mental picture of what the paper proposes, instead of a real-life application..
> - **Re: related works.** We initially placed the Background and Related Work section after the introduction, but moved it to the end of the work, as has become common in ML works. As the reviewer’s concern indicates, this change would have warranted adding more **citations to the introduction**, and we will do so (and move Section 6 forward) in a revision. Irrespective of the placement, we want to assure the reviewer that our work engages deeply with the literature. We closely surveyed several related areas, including algorithmic fairness, anti-discrimination law, multi-objective optimization, and machine learning. The gap we address results from this survey. We also do our best to highlight the most closely related work on LDAs and hope that our discussion of them in Sections 2 and 6 contextualize our contributions relative to them.
> - **Leaving Empirical Validation for Future Work.** This is a valuable question. We hope that the points at the top of this rebuttal address this in part. To add to it, we make the following clarifications. (1) Our main contribution is the reframing of the LDA problem in low-resource/information settings and ensuing theoretical scaling law. We hope that the reviewer recognizes this as an important contribution, as there previous works *entirely on the empirics* of scaling laws and *entirely on the theory* of scaling laws that precede ours [13-16]. (2) We agree with the reviewer that an extensive real-world study is important to viability, and we are already taking steps to engage with employment agencies to apply this framework to a real dataset. We believe this warrants a standalone follow-up work. (3) We want to note that, even so, we believe our synthetic experiments are important in that the main “gap” in our theoretical result is whether the assumptions hold. So we stress-test the assumptions in Section 5 via the synthetic experiments that test whether the theoretical result holds under a different data generating process than assumed, whether implicit regularization using training data manipulation is a good approximation of actual training, and if Assumption 4.2 holds in practice. Although we discuss this in the Appendix (E.2), we understand this should be clearer and will update Section 5 to reflect this.
> - **Model class being symmetric w.r.t. DGP.** It means that the model class $\mathcal{F}$ is not biased/favorable towards any particular distribution. Although this assumption may not hold exactly in practice, it is reasonable for large deep learning models, because they are meant to be universal function approximators. That is, as a model class, they are designed to be relatively setting agnostic and over-parameterized such that they could be symmetric wrt the DGP.
> - **Description of constants.** Thank you for this. To potentially overexplain, constants (that may depend on complicated quantities like $p$) are a standard tool used across, e.g., statistical learning theory (see, e.g., Prop 2.5 or Thm 2.6 of [17]). There are two main reasons they come in handy. First, the exact value of the constant is context-specific! Providing a specific value would not be meaningful. Instead, saying that the some variable Y (in our case, loss) depends on an independent variable X (in our case, fairness gap) according to our closed-form equation, where there are constants that need to be **fit** based on data is a traditional approach. Second, one could potentially provide a specific form for each constant, but that would require significant assumptions on the specific context. With respect to the scaling law literature, constants appear naturally to show that the scaling law takes a particular form, where certain constants must be “fit” based on data (e.g., see Equation 4.1 of [14]). Lastly, we note that one can trace the derivation of our constants in the proof on page 28 of the supplementary material.
> - **Line 305 varies with N.** Yes, the figure shown above Line 305 varies only N while keeping D constant.
>
> [1] Bertsimas et al. (2011). The price of fairness. Oper. Res.
>
> [2] Menon & Williamson (2018). The cost of fairness in binary classification. FAccT.
>
> [3] Kim et al. (2020). FACT: A diagnostic for group fairness trade-offs. ICML.
>
> [4] Navon et al. (2020). Learning the Pareto front with hypernetworks. arXiv.
>
> [5] Ruchte & Grabocka (2021). Scalable Pareto front approximation for deep multi-objective learning. ICDM.
>
> [6] Singh et al. (2021). A hybrid 2-stage neural optimization for Pareto front extraction. arXiv.
>
> [7] Rothblum & Yona (2021). Consider the alternatives: Navigating fairness-accuracy tradeoffs via disqualification. arXiv.
>
> [8] Kamani et al. (2021). Pareto efficient fairness in supervised learning: From extraction to tracing. arXiv.
>
> [9] Liu & Vicente (2022). Accuracy and fairness trade-offs in machine learning: A stochastic multi-objective approach. CMS.
>
> [10] Xu & Strohmer (2023). Fair Data Representation for Machine Learning at the Pareto Frontier. JMLR.
>
> [11] Liang et al. (2021). Algorithm design: A fairness-accuracy frontier. arXiv.
>
> [12] Gillis et al. (2024). Operationalizing the search for less discriminatory alternatives in fair lending. FAccT.
>
> [13] Hestness et al. (2017). Deep learning scaling is predictable, empirically. arXiv.
>
> [14] Kaplan et al. (2020). Scaling laws for neural language models. arXiv.
>
> [15] Hoffmann et al. (2022). Training compute-optimal large language models. arXiv.
>
> [16] Bahri et al. (2024). Explaining neural scaling laws. PNAS.
>
> [17] Wainwright MJ (2019). Basic tail and concentration bounds. In *High-Dimensional Statistics: A Non-Asymptotic Viewpoint*, pp. 21–57. Cambridge Univ. Press.

---

> > ### Comment · Reviewer_9GNb · 2025-08-03
> >
> > I thank the authors for their response. Having read what other reviewers had to say about the paper in their reviews and responses, there seems to be a consensus that what is holding the paper back is:
> > 1. Lack of clarity in theoretical results (i.e., constants)
> > 2. Lack of a demonstration
> >
> > However, I am not 100% confident that the authors have fully acknowledged and can effectively address these issues in time.
> > ### Re: Constants
> > I understand that the constants depend on the context and may have to be fitted. I don't think I asked for specific values. Rather, I was looking for intuitive descriptions in words. Considering this is a paper related to regulations in automated decision-making, those who are not well-versed in statistical learning theory may also pick up this paper.
> >
> > Nevertheless, I think I have a better understanding after taking a look at the supplemental material (I appreciate the pointer!).
> >
> > ### Re: Demonstration
> > I am not denying that exploring the LDA problem in low-resource/information settings is a valuable contribution. However, the authors precisely situate themselves in a very specific setting with a specific practical application in mind -- Title VII of the US Civil Rights Act and LDAs. As the authors have noted themselves, the "work is precisely motivated by a very specific, real use case." This is one of the main strengths of the paper (as reviewers, including myself, have noted in the reviews). However, this almost necessitates empirical demonstration of (1) the problem and (2) the solution proposed.
> >
> > Furthermore, many of the weaknesses pointed out by reviewers (e.g., constants) can be resolved by a step-by-step demonstration of the framework at play. Walking the readers through the process will clarify many details and illustrate its utility to a certain extent.
> >
> > I understand that access to a perfectly applicable dataset is very difficult. However, I still believe the authors could have been creative, even with synthetic data. The experiment design (and its analysis) is somewhat underwhelming as it stands.
> >
> > As a result, the paper is not as impactful as it can be (as noted by Reviewer **BDah**).

---

> > > ### Author Response · Authors · 2025-08-05
> > > **Author response**
> > >
> > > Thank you for engaging with our response and carefully considering the other threads. Below, we respond to your points.
> > >
> > > (1) **Re: constants.** We apologize that we didn't understand your meaning in the initial review, and we are glad that the pointers were helpful. As you may have perhaps deduced from the appendix, the constants are generally rates (or the produce of two rates), and the final constant $B(p, \mathcal{F}, \mathcal{D})$ refers to a term that can include both a constant shift and the general scaling law, i.e., $B(p, \mathcal{F}, \mathcal{D}) = \text{constant shift} + (1/\sqrt{N} + 1/\sqrt{D})$, where $N$ is the number of parameters in the model and $D = | \mathcal{D}|$.
> > >
> > > In terms of specific revisions, we plan to add intuition behind these constants into our main text, right below the main result. Relatedly, we had viewed the reviewer's concern (even in the initial review) as an indication that our discussion of Theorem 4.2 was not complete enough. As such, we are planning to flesh it out more fully by (i) explaining the constants as mentioned above, (ii) explaining how Theorem 4.2 relieves the burden/how we envision it can help given what we know about LDAs.
> > >
> > > To provide a preview of the latter, we paste part of our response to Reviewer BDah, which states that the approximate back-and-forth that we expect our framework can help address looks something like:
> > >
> > > 1. Plaintiff requests access to the model M and training data D to be able to produce an LDA
> > > 2. Defendant denies requests to the model M and training data D on the grounds of this being proprietary information (potentially appealing to trade secrecy) and even that revealing the training data D violates user privacy.
> > > 3. If the plaintiff wishes to use our procedure, they would express that, in the absence of M and D, the defendant should provide minimal information about the "type" of model used, the approximate model size N_target, the approximate training data size D_target, and a small portion of anonymized training data D' (even 1% to 5% of the train/test data would do).
> > > 4. If the defendant and judge agree, then the plaintiff runs our procedure, which is ...
> > > 5. Plaintiff trains small models of varying size on different fractions of D' and use this to fit the scaling law using Theorem 4.2. (Although this is a well known procedure, perhaps we took this for granted, and we can absolutely state the procedure for fitting a scaling law, which is to fit the constants $C, C', C'', C'''$ and $C''''$, where $C, C', C''$ come directly from Theorem 4.2 and we can break down $\mathcal{B} = C''' + C''''(N^{-1/2} + D^{-1/2})$, which introduces $C'''$ and $C''''$. We discuss this briefly in Sections 4.3 and 5 but acknowledge was not laid out in a clear, step-by-step way.)
> > > 6. Once the scaling law has been fit, then the plaintiff plugs in N_target and D_target, which immediately gives the extrapolated Pareto frontier.
> > >
> > > --
> > >
> > > (2) **Re: demonstration.** We appreciate the reviewer's points here and largely agree. We'll respond with two points.
> > >
> > > (i) As mentioned above and in other responses, one of our main proposed changes is to lay out the step-by-step procedure via which Theorem 4.2 can be applied. We do not anticipate this will take a significant revision, as our plan is to add that procedure to the end of the main results section or to the experiments section. If the latter, we would then modify to be titled "Demonstration and Simulations," where we would incorporate more of the discussion in the Appendix into the main text and explain how to fit a scaling law using Theorem 4.2 in a step-by-step procedure that largely re-iterates how we ran our simulations.
> > >
> > > (ii) Separately, the reviewer indicates an interest in us providing more creative simulations. We are happy to do so - we have actually run this procedure on other fairness datasets, like the Diabetes and COMPAS datasets. Due to early feedback we received, we pivoted to present results on synthetic data, as it allows for greater control over the setting/data generating process, giving results that more cleanly test the strength of the theoretical result/setup. However, given your feedback, we're happy to incorporate the appropriate results on those and similar datasets into a revision.
> > >
> > > We would like to reiterate/emphasize that our work (i) casts a seemingly impossible problem of establishing LDAs with limited model/data/compute access as a problem of extrapolating the PF; (ii) it's not clear that this problem is even solvable, and we show not only that the idea of scaling laws can be applied, but that more generally a *closed-form* applies (no such result exists in the literature). While we agree that experiments are highly important, we want to point out that in the absence of (ii), significant empirical studies are without a doubt necessary to show that scaling law-esque behavior would even hold across different datasets, settings, etc., but Thm 4.2 is a stronger affirmation of this.

---

> > > > ### Comment · Reviewer_9GNb · 2025-08-08
> > > >
> > > > I thank the authors for their thorough response! I will raise my score as the authors have addressed most of my concerns.

---

### Decision · Program_Chairs · 2025-09-17

**Decision:**

Accept (poster)

**Comment:**

This paper addresses the fact that while, in practice, individuals claiming discrimination could do so by demonstrating the existence of a "less discriminatory alternative (LDA)", this might not be feasible for regular people to do in practice. The paper then derives scaling laws for Fairness Pareto frontiers as a theoretical framework for solving the problem of discovering such LDAs.

Strengths:
- A compelling motivation and new, interesting problem -- this is why I have nominated the paper for a spotlight
- A nice mathematical formulation of the problem

Weaknesses:
- Lack of empirical validation on real-world examples. There is a lot of discussion back and forth about this between the authors and reviewers, and to be honest, while the reviewers were happy with the explanations, there is no way this can all fit into the paper. Thus, we must assume that the concern remains.

Recommendation:
I am recommending acceptance but the lack of real-world empirical validation is a major concern. If the paper is, indeed, accepted, the authors should make a major effort to include the most important parts of their discussions into the paper.